# The influence of subglacial lake discharge on Thwaites Glacier ice-shelf melting and grounding-line retreat

N. Gourmelen [1,2] ✉, L. Jakob [2], P. R. Holland [3], P. Dutrieux [3], D. Goldberg[1], S. Bevan [4], A. Luckman [4] & G. Malczyk[1]

The retreat of the Antarctic Ice Sheet is conventionally attributed to increased ocean melting of ice shelves, potentially enhanced by internal instability from grounding lines near retrograde bed slopes. Ocean melting is enhanced by increased intrusion of modified Circumpolar Deep Water (mCDW) into ice shelf cavities. Upwelling from the release of subglacial meltwater can enhance mCDW's melting ability, though its efficacy is not well understood and is not represented in current ice sheet loss projections. Here we quantify this process during an exceptional subglacial lake drainage event under Thwaites Glacier. We found that the buoyant plume from the subglacial discharge temporarily doubled the rate of ocean melting under Thwaites, thinning the ice shelf. These events likely contributed to Thwaites' rapid thinning and grounding line retreat during that period. However, simulations and observations indicate that a steady subglacial water release would more efficiently enhance basal melt rates at Thwaites, with melt rate increasing like the square root of the subglacial discharge. Thus, it remains unclear whether increased subglacial flooding events provide a stabilizing influence on West Antarctic ice loss by reducing the impact of subglacial water on ocean melting, or a destabilizing influence by triggering rapid changes at the grounding zone.

The Antarctic Ice Sheet (AIS) has contributed a total of $7.4 \pm 1.5$ mm to sea level rise between 1992 and 2020, with more than a twofold increase in mass loss rate during that period[1]. The AIS has the potential to significantly impact rates of future sea level rise. While sea level contribution from the AIS is projected to continue increasing, the magnitude of the future increase is still poorly constrained because of uncertainty in the ice sheet response to climate forcing[2–4].

The Amundsen Sea Sector of the West Antarctic Ice Sheet (WAIS) experiences the largest imbalance, accounting for the vast majority of the current AIS mass loss[1,5]. Mass loss is mainly realised via acceleration of the flow of ice into the ocean, whilst variability in surface mass balance plays a minor role[5,6]. In the last decades, ice shelves in the Amundsen Sea Sector have thinned, retreated, accelerated and weakened[7–9], and grounding lines have

retreated[10]. This took place in regions where grounded ice is highly sensitive to ice shelf and grounding line change, thereby leading to accelerated discharge of grounded ice into the ocean[11,12]. These changes have been driven by enhanced incursion of modified Circumpolar Deep Water (mCDW) onto the continental shelf, increasing ocean melting of ice shelves in the Amundsen Sea Sector, through some combination of a gradual anthropogenic warming during the 20th century and historical warm anomalies triggering ongoing change[13,14]. Due to the topography of the bed under the WAIS, and the Amundsen Sea Sector in particular, current changes may reflect an ongoing Marine Ice Sheet Instability and self-sustained, irreversible, retreat[15–20]. Such a collapse would have the potential to contribute several meters to sea level rise[21,22]. Within the Amundsen Sea sector, Thwaites

[1]School of GeoSciences, University of Edinburgh, Edinburgh, UK. [2]Earthwave Ltd, Edinburgh, UK. [3]British Antarctic Survey, Cambridge, UK. [4]Department of Geography, Faculty of Science and Engineering, Swansea University, Swansea, UK. ✉e-mail: noel.gourmelen@ed.ac.uk

Glacier is of particular concern due to the magnitude of ongoing changes and its potential to contribute significantly to future sea level rise[17].

The release of buoyant subglacial fresh water at the grounding line has the potential to alter ocean stratification and circulation, by creating turbulent upwelling of warm deeper waters along the ice-ocean interface and thus increasing melting[23]. In Greenland, where surface melt water production from atmospheric warming contributes significant quantities of melt water to the subglacial system, this process is thought to have led to substantial increase in submarine melting of tidewater glaciers[24–26]. Despite indirect evidence of enhanced melting by subglacial activity in Antarctica[27–30], and ocean observation of the presence of episodic subglacial water at the Thwaites grounding zone[31], there is a scarcity of evidence and no consensus on the efficacy of this process in impacting ice shelf melting, and on its role on the future evolution of the Antarctic ice sheet[32–35].

Here we analyse change in ice elevation, ice velocity, ice shelf basal melt rate, and ocean conditions at the margin of Thwaites Glacier and within the wider Amundsen Sea sector during the discharge of subglacial water from an extensive network of lakes located under Thwaites Glacier in 2013. We pair these observations with modelling of the ice-ocean-subglacial system and of the sensitivity of grounded ice to ice-shelf and grounding-line change to assess the extent to which subglacial discharge impacts the rate of ocean melting and the stability of grounded ice at Thwaites Glacier.

## Results

### Subglacial discharge
Satellite-derived ice sheet elevation change (see Methods) reveals that in early 2013 an extensive network of seven subglacial lakes, located between 44 km and 408 km upstream of the grounding line, started to drain under the Thwaites Glacier (Figs. 1a and S1) via a channelised system[36]. Routing modelling[37] suggests that all seven lakes are linked to a same connected hydrological system, exiting the ice sheet at the western end of an embayment in the grounding line of the Thwaites Western Ice Tongue (TWIT) (Fig. 1a)[38]. Lake drainage lasted about a year and discharged a total volume of $7.22 \pm 0.26$ km$^3$, about four times the predicted annual melt water production in this catchment, and the largest connected lake activity in the recent record under the AIS. The peak discharge occurred between late August and early September 2013 and reached 630 m$^3$ s$^{-1}$ (Fig. 2d), an eightfold increase from the steady-state discharge[36,39] and of comparable magnitude to summer peak meltwater discharge of the largest catchments of the Greenland Ice Sheet[24]. There was no evidence of elevation gain which would have indicated water storage downstream, hence subglacial meltwater from the lakes presumably was released to the ocean through the grounding line downstream[36,39].

A second episode of lake discharge took place in 2017 at the two most upstream lakes of the main branch of the subglacial network, Thw170 and Thw142[39]. During this event, there is surface inflation above the lake immediately downstream, i.e., Thw124, coupled with no noticeable activity further downstream in the network (Fig. S1)[39]. For the period following drainage, rates of lake recharge matches the expected background meltwater production[37]. As such we expect the discharge of subglacial water through the grounding line to have been largely suppressed after the 2013 event.

### Basal melt and ocean conditions
Satellite-derived oceanic melt rates (see Methods) show that basal melt rates under Thwaites fluctuate during the study period (Fig. 2e). In 2013, Thwaites experienced a period of transient increase in melt rate during which mean melt rates under the Thwaites Eastern Ice Shelf (TEIS) and near the grounding line of Thwaites Western Ice Tongue (TWITgl) nearly double (Figs. 2e and S2), from $17.5 \pm 0.7$ m yr$^{-1}$ to $31.0 \pm 0.7$ m yr$^{-1}$, with melt rates peaking in November 2013. The pulse

in basal melt rate is observed under both TEIS and TWITgl (Fig. 1), but is particularly pronounced in the latter sector where between 2013 and 2014 an additional $22.8 \pm 2.0$ m (or 8.1 Gt) of melt was generated (Figs. 1a and S3), from a baseline melt rate of 60.5 m yr$^{-1}$ (Fig. 1b), a substantially higher baseline melt than the melt rate under TEIS owing in part to a deeper ice draft (Fig. S4). During the same period, TEIS saw an additional $4.7 \pm 1.2$ m (or 5.9 Gt) of basal melt from a baseline melt rate of 7.0 m yr$^{-1}$. No such pulse in basal melt is observed under the Pine Island Glacier (Fig. 2e). Our melt rates are in close agreement with a recent high-resolution study[40] (see Methods). During the melt pulse event, Thwaites ice surface lowers rapidly by 3 m on average over TWITgl, and 1 m over TEIS (Figs. 2b and S5). TEIS does not appear to recover after the transient melt event where surface elevation remains lower than it would have been based on the 2011–2013 rates of change (Fig. S6). Over TWITgl a period of relatively lower thinning and melt rate takes place in the years following the melt pulse so that by 2018 surface elevation is at the level it would have been assuming constant 2011–2013 thinning rates (Fig. S6).

Basal melting is typically thought to be driven by ocean heat content (OHC) and circulation under ice shelves[41–43]. Two hydrographic moorings located in front of the Pine Island ice shelf, and indicative of oceanic variation impacting both Pine Island and Thwaites ice shelf cavities[44–46], show that the thermocline depth, a proxy for OHC, deepens from 600 m to 750 m between 2011 and 2013 (Fig. S7), and oscillates around a depth of 700 m during the following five years (Figs. 2e and S7). We found no significant correlation between the thermocline depth and basal melt rates under Thwaites and Pine Island across the entire study period (Fig. 3a). This lack of correlation suggests that the relationship between melting and OHC is not straightforward, or is not well represented between our moorings and satellite melt derivation (Methods). Geometric feedback and ocean circulation, on the continental shelf but also at the ice front and inside the cavity, or strong ocean stratification, can modify this relationship[31,33,47]. Basal melt rates under the Thwaites and the Pine Island ice shelf cavities are weakly correlated with each other over the entire period (Fig. 3a), driven by the large 2013 transient increase in melt rate under Thwaites which is not observed under Pine Island (excluding this event, correlation reaches 0.7).

In early September 2013, a polynya started forming about 20 km offshore from Thwaites' grounding line (Figs. 1a and S8), reaching a maximum extent of 86 km$^2$ by November 2013. The polynya is embedded within undisturbed sea-ice and icebergs and as such differs in both size, aspect, and timing from large wind-driven polynyas frequently occurring in the sector, but resembles sensible-heat polynyas associated with subsurface ice-shelf outflows, which bring warm deep waters to the surface[48,49]. This polynya is unique in the recent observational record at Thwaites Glacier (Fig. S8) and develops during the peak of subglacial discharge, slightly preceding the peak in basal melt rate increase under Thwaites. Thus we speculate that the polynya is the consequence of the buoyant plume created by the subglacial discharge and meltwater, entraining deep ocean heat, delivering it to the ocean surface, and melting the sea-ice.

### Rapid grounded ice thinning and grounding line retreat
Between 2014 and 2017, the western side of an embayment in the TWIT grounding line experienced a combination of strong localised grounded ice thinning, flow acceleration, and grounding line retreat[38,50,51] (Fig. S9). During this period surface lowering of up to 80 m and rates of basal melt of up to 200 m yr$^{-1}$ have been reported[50,51]. Although higher rates of grounding line retreat have been recorded elsewhere along the grounding line of Thwaites Glacier, including during this study period, the magnitude of basal melting and thinning was unprecedented during the satellite record[50].

The location of the ice thinning and grounding line retreat coincides with the location of both the downstream termination of the

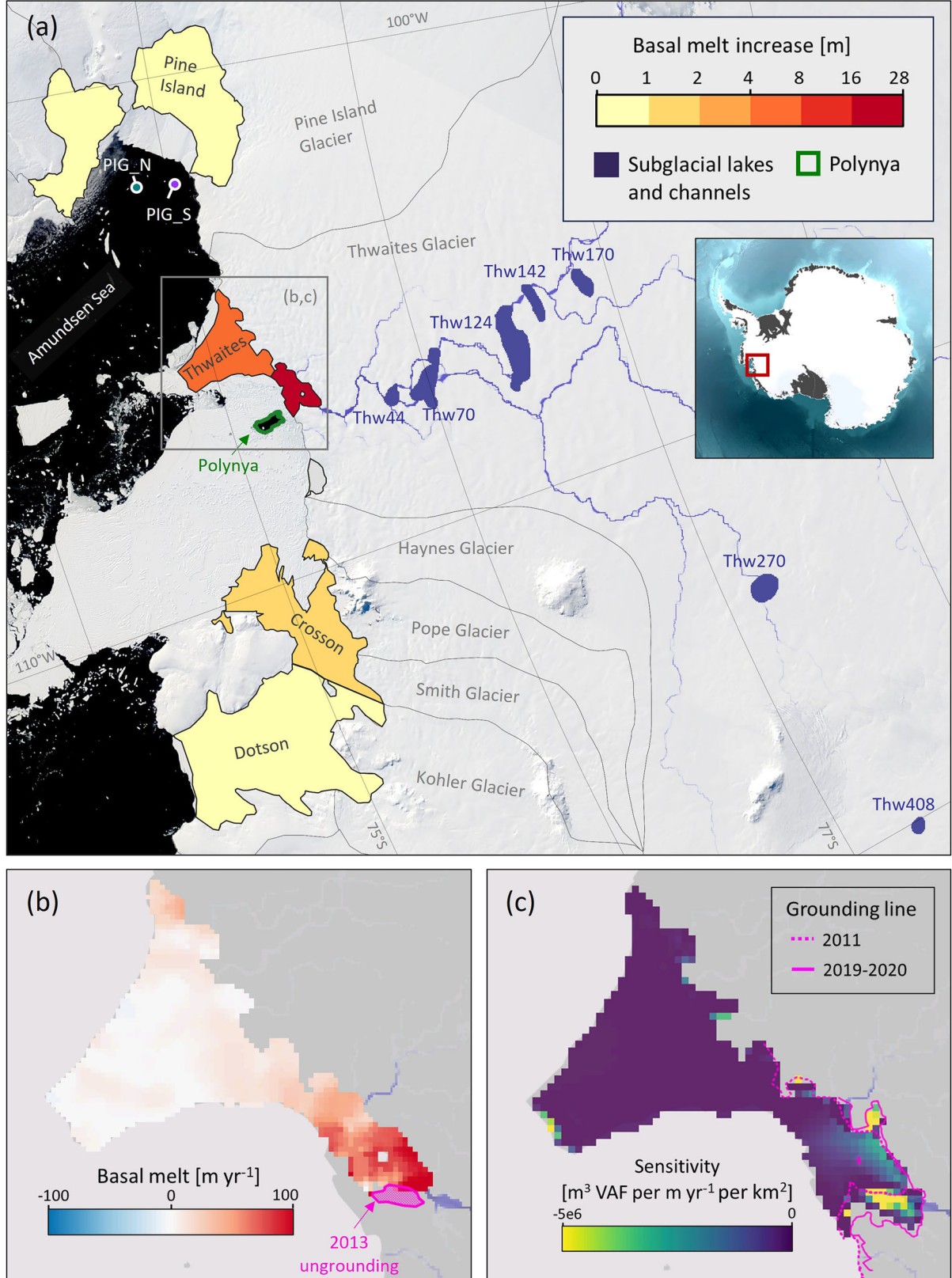

main subglacial network linking the active subglacial lakes and the area of the most intense anomaly in basal melting (Fig. 1a). Our observations of satellite-derived ice sheet elevation changes (Fig. 2c) show that, prior to thinning and retreat, the area was lowering at an average of 4.4 m yr$^{-1}$, consistent with the general rates of dynamic thinning reported at the ice sheet margins[52,53]. From approximately April 2013

the thinning accelerated, with the height of the ice suddenly lowering at rates of 11.7 m yr$^{-1}$. From 2017 onward, rates of surface lowering dropped to 1.2 m yr$^{-1}$ indicating that the area is now largely ungrounded.

We note that changes in geometry led to enhanced basal melt under Thwaites via "geometrical feedback"[33], particularly due to the

**Fig. 1 | Map of the Amundsen Sea Sector showing ice shelf basal melting and the active subglacial lakes network. a** Map showing the location of ice drainage basins, ice shelves, Thwaites's subglacial hydrology, polynya, and hydrographic moorings in front of Pine Island ice shelf. Basemap is a MODIS image acquired on the 5th of January 2014 during the polynya formation. The ice shelf colours display the amplitude of basal melt change during 2013–2014 coincident with the basal melt pulse at Thwaites, the ice shelf extent corresponds to the minimum for the 2011–2021 period accounting for calving front and grounding line variation[36]. Inset shows sector location within the Antarctic Ice Sheet (**b**) Mean basal melt rate under Thwaites Eastern Ice Shelf (TEIS) and near the grounding line of Thwaites Western Ice Tongue (TWITgl) between 2010 and 2020, the area of thinning and retreat at the grounding line is labelled "2013 ungrounding"; **c** Sensitivity of the grounded ice to changes in the Thwaites floating ice (see Methods)[64]. Source data are provided as a Source Data file.

onset of melting under the areas newly exposed to the ocean and due to changes in ice basal slope. This melt increase took place progressively from 2011 to 2017, with a higher melt increase between early 2014 and late 2015 reflecting the rate of grounding line migration along the western side of the grounding line embayment at TWIT[33].

## Discussion

### Link between basal melt pulse, lake discharge, and grounding-line thinning and retreat

Subglacial discharge, increase in ice shelf ocean melting, and grounding line thinning and retreat all took place simultaneously (Fig. 2a). This raises the question of the connection between these seemingly distinct processes, as well as the source of the perturbations.

It is plausible that grounding line thinning and retreat, by locally changing the ice surface, hence the hydraulic potential, and shortening the grounded ice between the lakes and the grounding line, would lead to a change within the subglacial system and act as a trigger for lake discharge[54–56]. If this were the case, we might expect lakes nearer to the grounding line to be impacted first, with lake activity progressively migrating upstream. Although there is uncertainty in the relative timing of drainage[36,39] all evidence points instead towards a cascading drainage with the most upstream lakes discharging first and initiating a cascade of lake drainage propagating downstream[39]. We also note that the two most upstream lakes Thw170 and Thw142 experienced a second episode of drainage in 2017 with no apparent grounding line change and with the lower part of the subglacial system seemingly shut down, suggesting that the subglacial system under Thwaites can be activated by triggering mechanisms unrelated to grounding-line change. Furthermore, most of the rapid grounding line retreat and thinning takes place further downstream of the modelled location of subglacial outflow through the grounding line (Fig. 1b), hence having a relatively small direct impact on the hydropotential over the drainage network itself.

The trigger for rapid grounding line thinning and retreat is also unclear. While ocean warming has been advocated[50], the 2013 events take place during relatively cold oceanic conditions within the wider Amundsen Sea sector[31,42,44,57,58] (Fig. S7), although a moderate transient increase in heat content takes place from early 2013 to early 2014 across the Amundsen Sea (Fig. 2e). The ungrounding of a pinning point immediately downstream of the grounding line could also provide a new pathway for warm water to reach the grounding line and reduce ice shelf buttressing[33,40,50]. However, ungrounding takes place during the preceding year in 2012 and the increase in basal melting associated with this event is moderate, and is dominated by melting of previously grounded ice[33]. Finally, the geometry at the location of the 2013 grounding line thinning and retreat is not favourable to grounding line instability[50].

Lake discharge is also a potential trigger for the changes observed at the grounding line and under TWITgl and TEIS. The density contrast between fresh subglacial water and the ocean can generate buoyant plumes, increasing turbulence and entraining warm bottom water. This process, which can lead to increases in ocean melting along glacier faces and under ice shelves[23], is recognized as a significant component of frontal ablation around Greenland[24]. While subglacial discharge fluxes are relatively low around Antarctica if steady discharge is assumed[27], the release of water during events such as the 2013 lake drainage is comparable with the water volume mobilised during Greenlandic summer melting, which contributes significantly to glacier ablation[24]. We observe a significant correlation between the timeseries of subglacial flux and the basal melting under Thwaites, driven by the 2013 changes (Fig. 3a). Correlation between subglacial flux and basal melting is increased even further by considering a -three-month time-lag (Figs. S10 and 3b). The nature of the three-month lag is unclear but appears to be related to the differing length of the pulse in subglacial flux and basal melting, while the start of basal melt increase and subglacial flux release appear to be synchronous, the peak in basal melting takes place about two months after the peak in subglacial flux, and basal melting takes longer to return to its baseline value following the end of lake activity.

To test the influence of subglacial lake drainage, we conducted a set of simulations of ice-ocean-subglacial interaction using the MITgcm ocean model (see Methods and ref. 33. for a full model description). These simulations use an existing high-resolution model of the cavity beneath TWIT, which included subglacial discharge at the steady-state rate, assuming no flooding events[33]. Here we use a Digital Elevation Model from 2013 for the TWIT geometry, and then run a set of steady-state simulations with different values of the subglacial flux, which is input at the deepest part of the Thwaites grounding line only (Fig. S11d). These simulations suggest that the 2013 lake discharge could have caused an increase in mean basal melt rate of 11 Gt yr$^{-1}$, a 70% increase, under Thwaites glacier, and hundreds of metres per year additional melt rate along the TWIT grounding line (Figs. 4 and S11). These values are of the same orders of magnitude as the melt rate increase observed during the grounding line retreat[50]. Our simulations suggest a power-law relationship of exponent 0.54 between subglacial flux and ocean melting (Fig. 4). This relationship is consistent with our observations, where we found a power-law relationship of exponent 0.52 between subglacial flux and ocean melting (Fig. 3b). While the relationship between subglacial flux and ocean melting has been observed once at tidewater glaciers[59], we believe that this is the first time that this relationship has been observed for ice shelves. Our melt exponent sits within the wide range found in the literature, typically from 1/3 to 2/3[24–26,60]. The exponent is impacted by ice-shelf and outflow geometry and Earth's rotation, highlighting the complexity of parameterising the subglacial effect on basal melting in ice sheet models[60].

Subglacial discharge can also have a non-local impact on nearby ice shelves via modification of ocean flow and stratification[61]. We speculate that this process could account for the changes being observed further from the outflow region, under TEIS (Fig. 1). The presence of the polynya, forming at the time of peak lake discharge, also suggests a localised increase in ocean temperature at shallow depths, also observed in our model results (Fig. S11i–l), compatible with plume upwelling (Fig. S11).

### Implications for the stability of Thwaites Glacier

In this region, ice thinning, grounding line retreat, and possibly changes to the bed friction from the modification of subglacial water pressure[62–65], are all potential triggers for an increase in grounded ice discharge as this is the key sector buttressing Thwaites Glacier. We examine in Fig. 1c a proxy for spatially resolved ice-shelf buttressing:

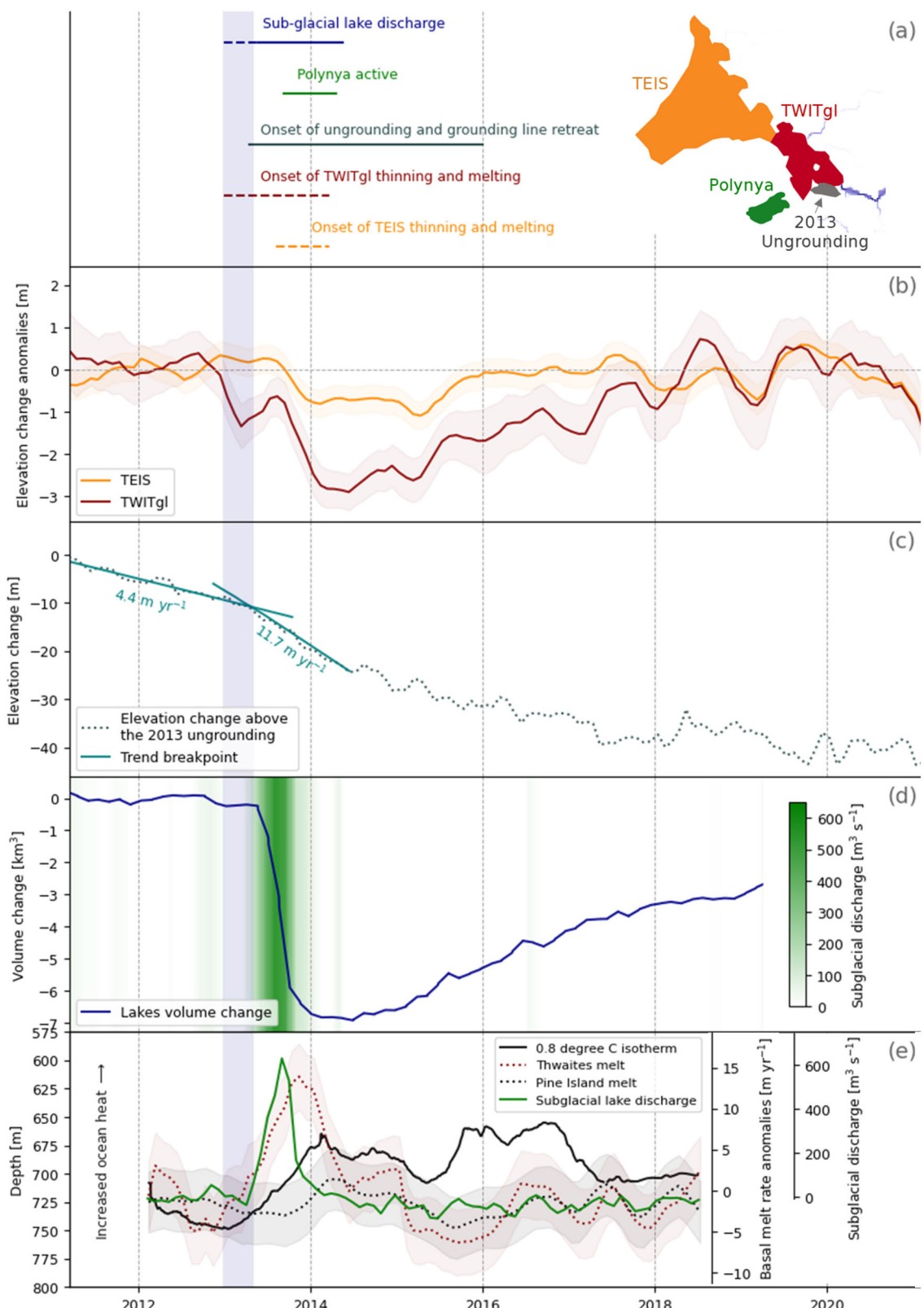

the sensitivity of grounded ice volume changes to the pattern of melt[64,66,67] (see Methods). The result shows that the largest impact of melt under floating ice is in the TWITgl region. The result agrees qualitatively with a different buttressing proxy based on the ice-shelf stress state[65], and together with the even stronger response under the 2013 ungrounding region, suggests that any increases to melt rates in this region will lead to a strong response in ice-stream thinning and

acceleration (Fig. 5)[64,65]. In the remainder we discuss impacts subglacial discharge may generally have on Thwaites.

The principal mode of subglacial water delivery to Antarctica's ice shelf cavities is still poorly known. While the sedimentary record suggests a preferential mode of episodic subglacial discharge[68,69], events like the lake discharge of 2013 under Thwaites appear to be unique in the satellite altimetry record[70]. As Thwaites continues to accelerate

**Fig. 2 | Relative timing between lake discharge, ice shelf thinning and basal melt increase, and grounded ice thinning. a** timeline of events and location of main features, subglacial network at the proximity of the grounding line is shown in grey; **b** time-series of elevation change anomaly from the 2011–2020 mean rate of elevation change for the Thwaites Eastern Ice Shelf (TEIS) and near the grounding line of Thwaites Western Ice Tongue (TWITgl), the elevation change is shown in Fig. S6; **c** mean elevation change over the area of the 2013 ungrounding (dashed line) and regression analysis over the period of increased thinning (solid lines) the

breakpoint is determined to be April 2013 (see Methods); **d** Total volume change of the seven active subglacial lakes (curves) and equivalent rates of subglacial discharge (green shading); **e** Change in thermocline depth from two hydrographic moorings in front of the Pine Island glacier (Fig. 1a), and basal melt rate variability under Pine Island and Thwaites (average for TEIS and TWITgl) ice shelves during the Thwaites' lake drainage and grounding line thinning and retreat. The vertical lilac shading across all five plots marks the onset of lake drainage taking place

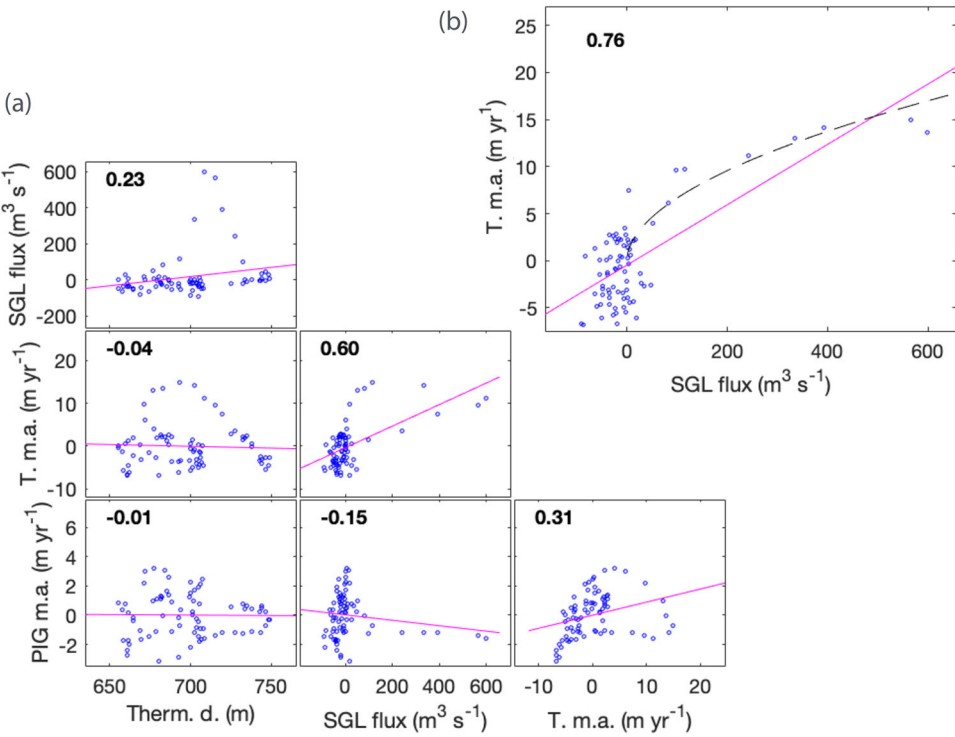

**Fig. 3 | Correlation between subglacial flux, ice shelf basal melting, and ocean conditions. a** Correlation between timeseries of Thermocline depth (Therm. D.), Subglacial flux (SGL flux), Thwaites basal melt anomaly (T. m.a.), and Pine Island Glacier basal melt anomaly (PIG m.a.) as shown in Fig. 2 of the main manuscript. Numbers in insets give the values of the Pearson correlation coefficients. $P$ values suggest that the Thwaites basal melt anomaly and the Subglacial flux timeseries (Pearson correlation coefficient of 0.60) have a significant correlation ($P$ value of $7e^{-9}$). Basal melt anomalies under PIG and Thwaites are weakly correlated over the

entire period and with a $P$ value of $6e^{-3}$. **b** A lag analysis suggests increased correlation between the subglacial flux and Thwaites' basal melt rates shifted backward in time by three months, the Pearson correlation increases from 0.60 to 0.77 with a $P$ value of $5e^{-16}$ (top right plot). Accounting for a three-month lag (Fig. S10), the relationship between Thwaites' basal melt and the subglacial flux is well approximated by a power law (dashed line) of exponent 0.52 (95% confidence interval [0.29 0.75]). Source data are provided as a Source Data file.

and thin, increasing both the supply of subglacial meltwater into the subglacial system as well as the hydropotential gradient of the region, it is possible that these rapid drainage events will become more frequent, and potentially activate currently stable lakes, the so-called "hydrologic catastrophes" beneath the ice sheet, delivering more subglacial water to the coastal oceans[54,71].

The mode of delivery of subglacial water, either as a steady flux or as episodic release via discharge and recharge of active subglacial lakes, impacts the efficacy of subglacial discharge in enhancing ocean melting under ice shelves. Ice-ocean-subglacial simulations suggest that melt rate enhancement can be approximated as the square root of the subglacial flux (Figs. 3b and 4). In the case of Thwaites Glacier, continuous steady release of the catchment subglacial water is modelled to increase ocean melting by 25%. In the case of episodic release via lake discharge of the same total volume once every eight years, it would only add an average of 9% more ocean melting each year (Fig. 4). Note this calculation assumes constant ocean conditions; episodic release coupled with variability in ocean conditions however may change the relative efficacy of episodic versus constant discharge.

The impact of a transient pulse of basal melt rate on the stability of an ice shelf and its grounding line is also poorly understood, and it is possible that in regions where the grounding line position is unstable a sudden pulse in ocean melting can trigger rapid self-sustained retreat. In the case of the 2013 events, the Thwaites' grounding line did not re-advance after the end of the transient melt increase. This took place in a sector where the bed is locally prograde[50] hence where the geometry does not in principle favour grounding line retreat. However, the increased melt rates induced by the geometric feedback[33] may also have contributed to maintaining the new grounding line position.

The location and strength of ocean melting under ice shelves determines its impact on ice shelf and ice sheet stability. The response of grounded ice is particularly sensitive to change in basal melt rates at the grounding line and along shear margins[64,66]. Subglacial discharge will impact the spatial distribution of basal melting under ice shelves. It will, to some extent, mirror and amplify background ocean circulation as both ocean-sourced plumes and subglacial-sourced plumes will be more pronounced along steeper basal slopes found near grounding lines. Subglacial amplification of ocean melting will also act preferentially downstream of large subglacial networks. Such conditions

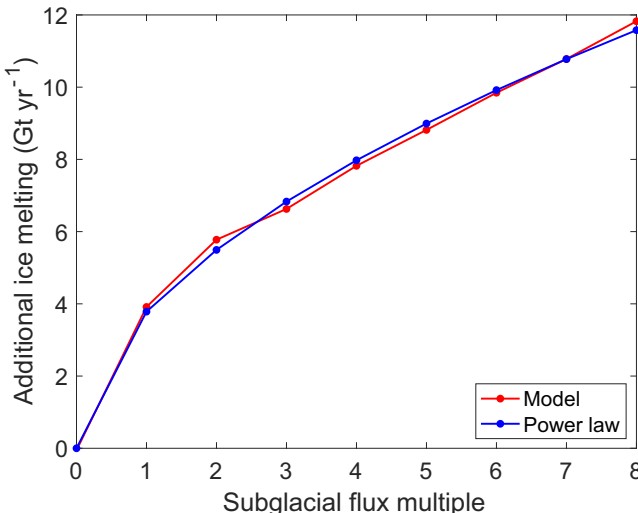

**Fig. 4 | Modelled additional ice shelf melt from subglacial discharge.** Predicted additional basal melt rate due to subglacial discharge (modelled: red; power law approximation ($\dot{m} = 3.78F^{0.54}$): blue; subglacial flux multiple $F$ is the subglacial flux used divided by 80 m³ s⁻¹, the background subglacial discharge rate for the main drainage network, subglacial flux multiple of 8 corresponds to the order of magnitude of the combined peak discharge of all seven Thwaites lakes in 2013. Source data are provided as a Source Data file.

are not limited to Thwaites Glacier, and are ubiquitous around the Antarctic margin, in particular at ice streams that drive the current Antarctic mass imbalance and will drive its future contribution to sea level[32]. Hence subglacial lake drainages, whilst infrequent, can likely play a role in modulating the ice response to ocean forcing, and therefore ought to be taken into account.

These observations highlight how little we still know about processes taking place at the grounding zone, despite its key importance for ice sheet stability. Amongst these processes, there is a growing body of evidence suggesting that seawater intrusions are taking place along Antarctic grounding lines, which may enhance the ability of the ocean to melt the ice sheet[72]. It is still unclear how these intrusions can interact with the subglacial system and the process described in our study. These two processes operate on different time scales, with seawater intrusions taking place via tidal modulations while subglacial lake discharge causing pulses of freshwater release at much lower temporal frequency. We can hypothesise that both processes interact, for example through cavities formed by seawater intrusions creating weakly grounded regions, enhancing the ability of subglacial discharge to cause sudden melt-driven ice retreat. Conversely, exceptional subglacial discharge could enhance seawater intrusion by flushing the grounding zone and melting out a wide sub-glacial connection that is then flooded by seawater. The process described here and the observation of seawater intrusions both challenge our current representation of grounding lines in the models used to forecast future ice loss.

## Methods
### Elevation and elevation change
**CryoSat-2.** Surface elevation change between 2010 and 2021 is derived from the CryoSat-2 radar altimeter using swath processing, a technique that allows retrieval of elevation measurements beyond the point of closest approach, increasing spatial coverage[73]. CryoSat's orbit and swath processing help to retrieve elevation at monthly intervals even over relatively small targets such as sub-glacial lakes (Fig. S12)[74]. Swath measurements are combined with a low-resolution conventional Point-Of-Closest-Approach using a Threshold first-maximum retracker (TFMRA) to minimize fluctuation in volume scattering[36,75,76]. The ratio of the number of swath measurements versus POCA measurements is

220 to 1, differences between swath and POCA elevation are low-pass filtered (100 km) to extract the large scale volume scattering signal, the difference is then subtracted from the full resolution swath dataset. Further consideration of volume scattering is discussed in the supplementary material, section 6c. Uncertainty follows the approach described in ref. 77, here we also add a term for uncertainty related to fluctuation in radar penetration, corresponding to 30 cm for each monthly elevation change estimate, following ref. 78. Volume change over the subglacial lakes (Fig. S1) is derived by integrating the elevation change over each of the seven lakes area displayed in Fig. 1a[37,39]. Figure 2d shows the volume change sum of all seven lakes. The change in elevation over ice shelves displayed in Fig. 2 corresponds to the mean elevation for the areas shown in Figs. 1 and 2, values for "Thwaites" corresponds to the sector including both TEIS and TWITgl. Elevation change anomalies correspond to the detrended change in elevation and are shown as elevation anomalies from the first two years of observation. To calculate the time evolution of surface elevation over the area of grounding line thinning and retreat (Fig. 2c) we consider the entire area affected by thinning between 2011 and 2018 in the high-resolution TanDEM-X analysis (Fig. S9). The breakpoint in thinning rate, i.e., April 2013, (Fig. 2c) is determined by running a 2-segments linear regression. To test for significance of the 2-segments model we also apply a 1-segment linear regression. *P*-values for the elevation time series are $4.6e^{-18}$, $3.4e^{-11}$, $1.8e^{-7}$ respectively for the raw time series and for the elevation (before and after the breakpoint) corrected with a 1-segment regression, indicating a significant elevation trend. Once corrected using the 2-segment regression, the dataset has a *p*-value of 0.27 indicating no significant trend.

**TanDEM-X.** High resolution elevation change used for the delineation of the area of grounding line thinning (Fig. S9) are derived from digital elevation models produced by the interferometric processing of TanDEM-X Co-registered Single-look Slant-range Complex images (CoSSCs) supplied by DLR. The processing was carried out using Gamma Remote Sensing software following the methodology described in ref. 51.

### Basal melt rate
We use mass conservation to derive ice shelf mean basal melt rate between 2010 and 2021, and basal melt time-series between 2011 and 2021, by combining measures of elevation and elevation change, Surface Mass Balance, firn air content, and ice velocity[8,74,75,79]. Melt time-series excludes data from 2010 and early 2011 due to more limited radar altimetry coverage. We update results obtained in ref. 75. over Thwaites by using 2015-averaged ice velocities from ITS_LIVE[80] as opposed to the ITS_LIVE composite as previously used. This allows retrieval of melt rate over a larger portion of the TWIT sector, impacted by fracture development in the ITS_LIVE composite. To calculate the melt time-series we consider variation in surface elevation, Surface Mass Balance and firn air content from the RACMO and FDM models[81,82] (Fig. S3). The impact of considering a constant ice velocity is discussed in the supplementary material section 10b and Figs. S8 and S12–S14. Note that the basal melt rate time-series in Fig. 2e is obtained from deriving the cumulative melt anomaly (Fig. S3) and is low-pass filtered with a smoothing window of 1 year to enhance the component of interest of the signal. Derivation of the monthly cumulative melt anomaly enhances noise in the data hence we derive monthly basal melt rate for the entire Thwaites sector, which reduces noise, but shows cumulative melt anomaly for all the sectors discussed (Fig. S3). Uncertainty in basal melt rates accounts for uncertainty in each of the mass conservation terms, for SMB and firn air content an uncertainty of 10% is applied, following refs. 8,75. Non-hydrostatic equilibrium could be affecting regions classed as ice shelves but at close proximity to the grounding line and pinning points[83], we therefore investigate the sensitivity of our result to non-hydrostatic equilibrium by excluding measurements acquired within 5 km of the

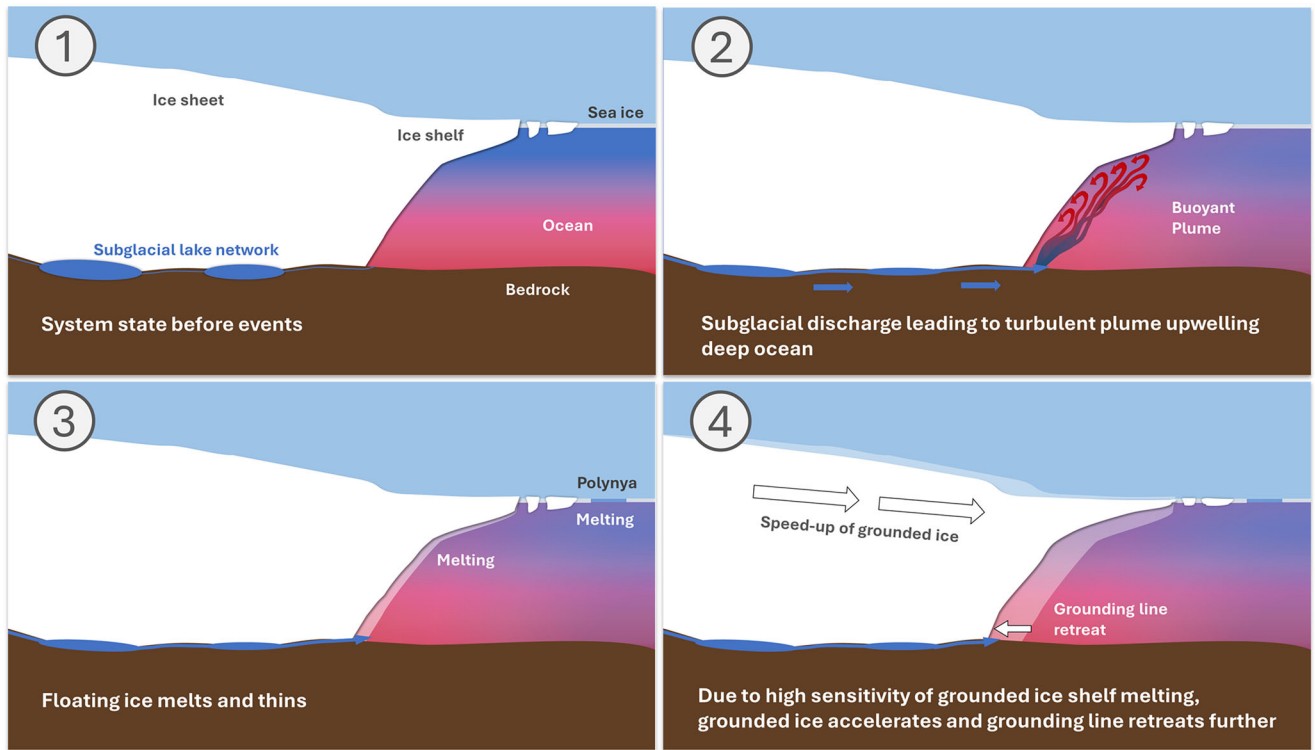

**Fig. 5 | Schematic of proposed succession of events during the 2013–2014 period under Thwaites Glacier.** Schematic diagram of impact of subglacial lake discharge on ocean melting and ice shelf, sea-ice, and grounding line change.

grounding line and pinning point, which shows insignificant impact on the final melt time-series (Fig. S6). We exclude from the calculation areas where a high shear rate indicates the presence of fractured ice that would potentially bias the surface elevation change and where the mass conservation approach breaks down[84]. This means that a large proportion of TWIT, beyond the TWITgl sector, is excluded from this analysis. Significant episodes of grounding line retreat have taken place in the Amundsen Sea region since 2010. To ensure that we only consider elevation measurements made over floating ice for the duration of the CryoSat-2 period, we use the 2011 grounding line definition[10]. Similarly, we consider the minimum ice shelf extent during the 2010–2021 period to ensure that melt variability is not affected by difference in spatial extent of ice shelves. However, this means that we do not capture melt in area that have ceased to be ice shelves, or that have become ice shelves, between 2010 and 2021[75]. The resultant mask means that changes in elevation and basal melt are reflective of the processes affecting ice that is in contact with the ocean for the duration of the observed period. Our melt rates agree well with a recent high-resolution melt study[40], mean basal melt rate of overlapping measurements is 20.4 m yr$^{-1}$ for this study, and 21.8 m yr$^{-1}$ and 21.6 m yr$^{-1}$ for ref. 40.'s 2011–2015 and 2016–2019 periods respectively. ref. 40. also found relatively higher rate of mean basal melting under TWITgl during a period including the basal melt pulse discussed in this study i.e., 66.9 m yr$^{-1}$ (2011–2015) compared with 52.5 m yr$^{-1}$ (2016–2019) and 55.7 m yr$^{-1}$ (2020–2023). The melt rate increase shown Fig. 1 and mentioned in the "basal melt and oceanic conditions" section is calculated from the basal melt anomaly (Fig. S3) and corresponds to the difference between the mean 2014–2015 melt and the mean 2012–2013 melt. The correlation shown in Fig. 3 is performed on all the data points shown in Fig. 2, for the computation of the power law shown in Fig. 3b we resample the melt data at regular intervals of subglacial flux i.e., from 0 and every 40 m$^3$ s$^{-1}$ in order to give equal weights to data points across the range of values of observed subglacial flux.

## Subglacial routing

The subglacial routing depicted in Fig. 1 is from ref. 37. The subglacial flow pathways are calculated following the thin-film-based subglacial water-flow method[37,85]. Subglacial drainage pathways are identified by first calculating the hydraulic potential using surface and bed topography from BedMachine Antarctica[86] and assuming basal water pressure is at overburden everywhere.

## Ice-ocean-subglacial modelling

We use the MITgcm model to simulate ocean currents and melting beneath the floating part of Thwaites Glacier, using the same model as ref. 33. We use a hydrostatic implementation of the MITgcm ocean model in a small 120 km × 120 km domain focussed on the ocean cavity beneath Thwaites Glacier (Fig. S11 shows a zoom on the region of interest). Wider ocean forcing is applied through restoring boundary conditions for temperature and salinity on the open-ocean boundaries. No ocean surface fluxes or sea-ice model are applied. The simulations rapidly attain a steady state in which heat and salt sources from the boundaries are balanced by the heat sink and freshwater source from ice-shelf melting. Simulations are run for six months and results averaged over the final month. The equations are solved on a 200 m (horizontal) by 10 m (vertical) grid, with a timestep of 30 s. Ice-shelf melting and subglacial inflow are implemented as virtual heat and salt fluxes to avoid a rising sea surface. A standard 3-equation ice-shelf melting parameterisation is used, with parameter choices following ref. 87. Constant viscosities (1 m$^2$ s$^{-1}$ horizontal, 5 × 10$^{-4}$ m$^2$ s$^{-1}$ vertical) and diffusivities (0.1 m$^2$ s$^{-1}$ horizontal, 5 × 10$^{-5}$ m$^2$ s$^{-1}$ vertical) are used. The agreement of the model results with the sparse available data for Thwaites Glacier are discussed in ref. 33.

In the present study, we fix the ice geometry to use a Digital Elevation Model from 2013, and then run a set of steady state simulations with different values of the subglacial flux, which is input at the deepest part of the Thwaites grounding line (Fig. S11d). We run one simulation with no subglacial flux, then one with the mean steady state flux of 80 m$^3$ s$^{-1}$

(ref. 37), and then a set of further simulations using multiples of this flux, up to eight times larger, $640 \, m^3 \, s^{-1}$, which approximates the magnitude of the subglacial lake flood input in 2013. Increasing the subglacial flux drives stronger buoyancy-driven currents beneath the ice, increasing melting. We fit a power law curve to the additional melting under TWITgl caused by the subglacial flux (Fig. 4), with a best-fit exponent of 0.54 (95% confidence interval [0.48 0.60]). We chose TWITgl as this is the sector close to the outflow where the model is most sensitive to the impact of the subglacial flux, and is the region of highest melting, which impacts the main trunk of Thwaites. This demonstrates that in our modelling, the additional melt rate varies as the square root of the subglacial flux, implying that episodic flooding is less efficient at enhancing melt.

### Ocean observation

Moored observations between January 2009 and 2020 provide time-series of ocean temperatures between 350 m to the oceanic seabed in two different locations in Pine Island Bay, just East of the Thwaites ice shelf[88]. The 0.8 °C isotherm serves as a proxy for the thermocline depth separating relatively warm and salty mCDW near the seabed to the colder and fresher surface waters interacting with the atmosphere above. This isotherm depth can be tracked in both records, with clear co-variability over a five year's time period. While the moorings are missing the upper ocean, the upper ocean has limited to no access to the cavity. This is especially true for the deeper parts of the cavity, which should mainly be concerned with heat content >400 m depth. Whilst moored observations do miss the upper 350 m of the water column, the thermocline separating surface and bottom water can clearly be identified at all times. The OHC recorded by the mooring is therefore a good proxy for the glacial melt available OHC outside of the cavity. Gaps in the longer record are filled using the shorter record by removing a mean depth difference by least square fit, providing a 2009–2020 record of thermocline variability in Pine Island Bay.

### Ice sheet modelling

The sensitivity of grounded ice loss to melt rate on a grid-cell basis is estimated based on an ice-sheet model of the Amundsen initialised with a fixed ice surface, thickness and bed geometry, and flow parameters are calibrated using observed velocities[64]. The ice-sheet model is then evolved over a time period with imposed melt rates, and a loss of Volume Above Floatation (VAF) is calculated. Then, using Automatic Differentiation[67], the sensitivity of this VAF loss to perturbations to melt in each model grid cell is calculated. In other words, the result is a gridded product, such that its value in a grid cell is a linearised estimate of the *additional* VAF loss that would occur, if the melt rate were perturbed by 1 m/a in that cell only. For this study, the calculation was updated from that of ref. 64. in the following ways: the grid was more highly resolved (1.25 km rather than 1.5 km); the geometry and velocity product were updated (to BedMachine-Antarctica v3 and MEaSUREs v2, respectively); and only one ice-sheet model, STREAMICE (the ice-sheet component of MITgcm[67]) was used. The "baseline" melt was parameterised as varying linearly with depth, from 0 at the surface to 50 m/a at 700 m, and 50 m/a below this. A parameterised, rather than observed or modelled, melt pattern was used, as it meant that as the ice sheet ungrounded, it would still be exposed to melt.

### Data availability

CryoSat-2 data are available at cs2eo.org; Velocity data are available at https://nsidc.org/apps/itslive/, Surface and bed topography are available at https://nsidc.org/data/nsidc-0756/versions/3; Mooring data are available at https://www.seanoe.org/data/00887/99922/. TanDEM-X elevation change are available at https://doi.org/10.5285/DF8C4AC0-1723-43AE-AD48-D02D58699F32. MODIS images are from the World-view Snapshots application (https://wvs.earthdata.nasa.gov), part of the Earth Observing System Data and Information System (EOSDIS). The CATS tide model can be found at: https://www.esr.org/research/

polar-tide-models/list-of-polar-tide-models/cats2008/. Source data are provided with this paper. Source data are provided as a Source Data file, additional data are available in the following repository: https://doi.org/10.5281/zenodo.14774213. Source data are provided with this paper.

### Code availability

The MITgcm adjoint model was run using checkpoint 68 u of the MITgcm code (https://github.com/MITgcm/MITgcm). Automatic differentiation was done with the OpenAD differentiation engine (https://mitgcm.readthedocs.io/en/latest/autodiff/autodiff.html#adjoint-code-generation-using-openad). Codes to process CryoSat-2 data can be found here: https://git.ecdf.ed.ac.uk/cryosphere/thw-sgl-mlt. Codes to generate the results of this study and figures can be found here: https://doi.org/10.5281/zenodo.14774476 and here https://git.ecdf.ed.ac.uk/cryosphere/thw-sgl-mlt.

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

## Acknowledgements

The authors gratefully acknowledge the European Space Agency (ESA) for the provision of CryoSat-2 data, the National Aeronautics and Space Administration (NASA) for the provision of MODIS data. N.G. and D.G. were supported by ESA's 4DAntarctica's project (4000128611/19/I-DT). N.G. and L.J. were supported by ESA's Polar+ Ice Shelves project (ESA-IPL-POE-EF-cb-LE-2019-834), the SO-ICE project (ESA AO/1–10461/20/I-NB) part of the ESA Polar Science Cluster. N.G. and D.G. were supported by the PROPHET project and P.H. was supported by the MELT project, both components of the International Thwaites Glacier Collaboration (ITGC) with support from National Science Foundation (NSF: Grant #1739031) and Natural Environment Research Council (NERC: Grants NE/S006745/1, NE/S006796/1, NE/T001607/1, and NE/S006656/1). ITGC Contribution No. ITGC–143. Amundsen mooring observations are maintained by NERC/BAS under grant Ocean Forcing Ice Change NE/N062102. TanDEM-X data were provided by DLR through project GLAC0323.

## Author contributions

N.G. designed the study. N.G. and L.J. processed and analysed the ice elevation and basal melting data. P.H. generated and analysed the ocean modelling data. P.D. acquired and analysed the oceanic moorings data. D.G. performed the ice sheet sensitivity analysis. S.B. and A.L. processed the TanDEM-X data, G.M. processed the subglacial lake data. N.G. wrote the manuscript. L.J. prepared the figures. N.G., L.J., P.H., P.D. and D.G. contributed to scientific discussion, interpretation of the results, and writing of the manuscript. N.G., L.J., P.H., P.D., D.G., S.B., A.L. and G.M. reviewed the manuscript.

## Competing interests

The authors declare no competing interests.
