## [Transparent Peer Review file · Nature Communications]

The influence of subglacial lake discharge on Thwaites Glacier ice-shelf melting and grounding-line retreat

Corresponding Author: Professor Noel Gourmelen

Version 0:

Reviewer comments:

Reviewer #1

(Remarks to the Author)

The paper is in good shape and describes important results that will be of interest to the ice-ocean interactions community as well as more broadly for interpreting and predicting future ice loss. The work is interdisciplinary, drawing from ice and ocean observations and from modeling results. I find the paper nearly ready for acceptance, and would support publication after minor revisions.

The key results show a novel event in which enhanced ocean melting of the Thwaites Ice Trunk is likely driven by an anomalously protracted and voluminous subglacial outflow event. The evidence from surface elevation data, when combined with ocean data and compared to modeling of the event is convincing that the elevated subglacial discharge is indeed responsible for the observations. The work also notes the evidence for presumed outbursts in the geological record as a contributing factor to past evolution of ice sheets, while paradoxically the results of this work suggest they do not likely determine the overall trends in ice sheet retreat. That connection is very important.

There are two general issues with the presentation of the paper; these are mostly superficial and thus under revision the paper would be much improved. First, there is little transition between sections of the effort--while using remote sensing and ocean data, there is little context given for the observations and their novelty within the main text, which should be improved. The methods are a place for detailed techniques but the main paper should describe better what the context of the observations are and how they are interpreted, not just delve straight into the interpretation. Similarly, the modeling shows up with not descriptions or context. Some lead in to the simulations describing them and their initial set up and purpose would improve reading of the paper.

There are no significant flaws in the methods or interpretations. However, the last section of the paper trends toward speculation and could be improved with tightening of the logic and greater thought or direct commentary on whether any similar systems might exist within the observational record.

Towards the model descriptions and context, they are somewhat brief even in the methods. Further detail on the modeling would improve the paper. Based on the written description, it is hard to know whether the models would be trivial to reproduce.

Of minor but important note, several times in the paper a cavity that has recently been observed forming in the TWIT is referred to as the "Milillo Cavity" or similar; while it is a useful shorthand to describe features with names like this, especially in Antarctica where there is extreme sensitivity regarding how places are named, it would be better to find a different shorthand that is agnostic and not connected to a particular scientist. For example, the "new cavity" or "TWIT GL crescent" or something geometric or similar would be more appropriate (with all due respect to the work of Milillo et al who made an important discovery). Unofficial names with reference to individuals should be avoided in general. Since the subglacial ocean environment proximal to the GL is already called a cavity, perhaps a new description would be more useful.

I provided a commented document with tracked changes for other notes and comments on the main text of the paper, as well as a few comments below.

Minor comments:

Last sentence of abstract should be rewritten, as there is no evidence here or otherwise presented that would suggest that subglacial events would be stabilizing.

'These' in paragraph 1 (P1) is vague. Specify what is projected to increase, ie 'While SLR contributions..'

Beginning of P3, first sentence possibly needs a citation or several.

In the section on basal melt and ocean conditions, the description of surface elevation changes of TEIS and TWITgl at the end of the second paragraph is somewhat confusing. Looking at Figure S2, it is not clear that TWITgl recovers to where it would've been by 2018 and that TEIS doesn't recover. See comments for Figure S2 for suggestions to increase clarity.

Figures:

Figure 1a: Since each glacier is assigned one color it would make more sense to have a discrete colorbar rather than a continuous one. Continuous colorbar makes sense for panels b and c.

Figure 3: Consider swapping this figure with Figure S4. Also could use a description of what the 'power law' is in the caption.

Figure S2: It would be helpful to show on the plot where the 2011-2013 rate of change falls so the readers are able to compare the observed values to something and to include a line on both plots showing where the elevation change would have been if following the 2011-2013 trend.

Figure S3 would benefit for some annotation showing readers where they should be looking for the polyna and how it is different/stands out. The outline doesn't match the outline in Figure 1

Figure S4 caption be consistent with /s and s-1, and with the timescale of the simulation.

Figure S6 needs y-axis labels

Reviewer #2

(Remarks to the Author)

The manuscript examines the Thwaites area and uses elevation changes, melt rate estimates and some hydrographic timeseries to consider interaction between the ocean, ice shelves, and a network of subglacial streams and lakes. The region is certainly critical for projections of future sea level.

I struggled as the manuscript does not clearly set up what it is going to examine and what the driving questions are. The introductory section just stops and then key scene setting points are made later on. The introduction is followed by subsections examining separate aspects of the data. Then finally a subsection comes along that reveals the possible connections between the various elements.

What are the noteworthy results?

The key points seem to be - that there is interaction between the subglacial drainage and the shelf configuration and that none of this is particularly steady in time. To my mind the study doesn't establish the characteristic timescales for any of the processes. Instead, two months is described as quasi-simultaneous.

Will the work be of significance to the field and related fields? How does it compare to the established literature?

The region is certainly critical for projections of future sea level and there has been little work relating coastal processes to subglacial stream/lake transients that I'm aware of - I'm made a couple of reference suggestions.

Does the work support the conclusions and claims, or is additional evidence needed?

The manuscript leaves the conclusions rather open-ended.

Are there any flaws in the data analysis, interpretation and conclusions?

I can't see flaws in analysis but as above the interpretation leave things open.

Is the methodology sound?

Yes

Is there enough detail provided in the methods for the work to be reproduced?

At a broad level probably

I think the story would benefit greatly from a clear sketch showing the system, the processes and the "direction of travel of effect".

There could be a lot more introduction and clarity when referring to various regions. More generally, I would have preferred more commas.

There were no line or page numbers in my pdf so I marked the manuscript – unfortunately this seemed to result in a very large file size – apologies for that.

Reviewer #3

(Remarks to the Author)

Reviewer #4

(Remarks to the Author)

The paper by Gourmelen et al. describes a lake drainage event that occurred on the Thwaites Glacier in the Amundsen sector of West Antarctica. Although the methodological developments of the paper are not groundbreaking and are rather standard, this study highlights a unique event that could potentially impact the stability of the Thwaites Glacier, if it becomes more frequent in the future.

Overall, the paper is well structured and well written. Lines and page numbers should be added to facilitate the review process, right now it is really difficult to reference correctly the text in the review. There is still work to be done on the figures, which sometimes do not provide the necessary information to support the arguments made in the text (see comments below). Critical methodological information is also lacking at the moment (see specific comments below). Finally, the paper should better discuss recent findings by Rignot et al., 2024; which have highlighted significant grounding line modulation in this region that could challenge the proposed processes of this paper.

More importantly, the paper does not satisfy the open-science requirement of nature journals, and nature communications more specifically. This is currently a really concerning issue. There is not even a data availability/code sharing statement. There is no sharing of any of the results, and no sharing of any of the codes to retrieve the data. It is therefore not possible to verify or reproduce any of the results that are presented here, which is a major obstacle for the paper review. As a consequence, I would formally oppose the publication of this study, until the data necessary to verify the methodology and results have not been provided. This includes: - maps of basal melting rates throughout the entire study period, elevation maps both from Cryosat-2 swath data or POCA, and codes to replicate the processing performed by the authors (swath processing, retracking method), along with CSVs for all data used in the figures.

Specific comments

- The paper needs a small section on the methodological approach and the overall objective of the paper before running into the result section.
- Add error bars to the changes in elevations (Fig S1), this should also include penetration uncertainties
- Please precise how you calculated the channelized system
- How did you derive the changes in surface elevation? Over which area? Is this an average? Single point? Please add on figure 1 the area where you calculated the trends.
- How often do you have measurements of surface elevation, ie how confident are you with the timing of lake drainage? Please be more specific.
- Can you estimate radar penetration biases using ICESat-2 for confidence over 2018-present?
- Please be more specific of the routine modeling. How was this calculated? You cannot expect the readers to know everything, you have to provide a certain level of details.
- "During this event, all water...network." Which lake are you referring to? Can you see an elevation gain which would justify that everything has been collected there? Please provide more details.
- "Satellite-derived oceanic melt rates... 10.8 m/yr." Over which area did you calculate this average melting rate? Since a large part of Thwaites has collapsed it is of prime importance to understand where you calculate this change in melting rates.
- "near the grounding line (Figure 1a)". Are you referring to Figure 1b?
- "TWITgl is characterized by a deeper... average of TEIS." Please provide a figure to justify that statement.
- "Basal melt rates under Thwaites...double (Figure 2e)". Please also provide the basal melt rate time series instead of the anomaly as a supplementary figure. Right now it is difficult to see if it doubled or not. Figure 2e only shows the anomaly for what is called "thwaites" in general. So it is a bit unclear to see at what location the time series has been calculated. Please provide it also for the different subregions, in order to look at if we can observe the pulse everywhere.
- "The pulse in basal melt rate is observed...TEIS and TWITgl". I am a bit sceptic about the TEISgl and TEISfront zones. What criteria was used to differentiate both regions?
- "During the melt pulse event...(Figure 2b)." It is really difficult from the provided figures, to assess that these statements are convincing. 1) Figure 2 caption says it is cumulative elevation change anomalies, but the legend only says elevation anomaly, and the text describing figure 2b talks about elevation. Please make things clearer.
- "28 m on average over the TWITgl". I don't really see how this is being retrieved from figure 2b. At the end of the melt pulse event, I can see that the elevation change anomaly over TWITgl is -3 meters. How do you calculate these anomalies? With

respect to what year? Please provide further details.

- "TEIS does not appear to recover... rates of change (Figure S2)". From Figure 2e, the elevation change anomaly remains around zero for TEIS, so I don't really understand the fact that TEIS does not recover.
- "Over TWITgl... thinning rates (Figure S2)." Did the author switch TWITgl and TEIS? because based on this description this is not what Fig S2 is showing at all.
- "thermocline depth... in the following 5 years". The thermocline depth increased by how much?
- Figure 2e. The dotted lines are the basal melting rates, but the Thwaites melt label is placed right on the moorings. Please have a proper legend. Plus it is impossible to really see if PIG melt follows the thermocline depth. Indeed the basal melt anomaly is close to zero most of the time. We have a slightly positive anomaly in 2013, but after that, anomalies seem always negative while the thermocline depth continued to increase. After 2016, the thermocline depth decreased but it seems the only year when the PIG melt anomaly looks positive again. A larger graph on this specific period is needed to conclude.
- "Similarly, basal melt rate under Thwaites...". This is not obvious since the basal melt anomalies are negative or close to zero from Fig 2e
- "10 km by 5 km". Please provide an area of the polynia instead. This does not mean anything since the polynia is not a square.
- "Between 2014-2017". If you talk about the Milillo cavity as referred in the figure, please state the reference name here.
- "During this event, surface elevation...(Figure 2c)". This is not what Figure 2c is showing, please also provide time series of basal melting rate on top of anomalies.
- "4.4 m/yr". Please indicate the rates close to each of the fits that you show in Fig 2c.
- "While ocean warming has been advocated...". The lake volume change, shown in Fig 2d, is really coherent with basal melting increases and the ocean temperature increases. The exact timing of the event has an errorbar (return period of cryosat), which has to be considered and might explain why the peaks do not exactly correspond.
- "The ungrounding of a pinning point..." Are you talking about grounded ice here? Otherwise I don't understand how the unpinning of a pinning point will provide new warm water pathways, since this is mainly controlled by the bathymetry itself. The unpinning can however decrease the buttressing but has no effect on the warm water pathways. If authors have other thoughts in mind, please provide more details.
- First, note that it is possible to have grounding line retreat on prograde slopes too in case of instability (Hill et al., 2023). Furthermore, recent work from Rignot et al., 2024 have pointed toward significant grounding line modulation specifically in the Milillo cavity (up to 5 km inland). In this region, the authors have shown that irregular warm water intrusions are penetrating up to 6 km inland beneath grounded ice. Such phenomena could challenge your proposed explanation, given the uncertainty on the timing of lake drainage. Please discuss this point.
- "Implication for stability of Thwaites". It is not clear whether you are able or not to reproduce changes in melting rates below the ice shelves with the addition of freshwater.
- Can you more clearly express if you are able to reproduce the changes observed in runoff and basal melt rates using the model? I think you have all the tools to do that. Using a simplified approach, as in Rignot et al., (2016), which model submarine melting as a function of increased ocean forcing and runoff, could also be a good and more direct point of comparison. Please provide a scatterplot or comparison of modelled vs observed melting rates using the measured freshwater discharge.

Methods section

- How much swath vs POCA was used? What is the impact of the large imprint of POCA on the resolution of basal melting rates? How much is the CryoSat-2 penetration? Does it evolve with elevation/snow conditions?
- Please provide a code repository for CryoSat-2 processing as it is central to the paper findings, and to comply with open science/FAIR principles
- Basal melting rates: Provide the data used to derive the results of the study (time series of basal melting rates). It is critical to verify the analysis and to comply with open science/fair principles
- Basal melting rates: Provide details on how this was calculated. Which ice velocities product, did changes were monitored in lagrangian framework? If not, the final product of basal melting rates could be much more aliased (see Shean et al., 2019). Also, which firn air content products? Provide details for what you call "standard approaches". You can't expect people to know everything, the methodology should be reproducible, and it is not the case right now. What are the uncertainties on firn air content and SMB?

I am reminding below the guidelines for data sharing in nature journals.

A condition of publication in a Nature Portfolio journal is that authors are required to make materials, data, code, and associated protocols promptly available to readers without undue qualifications.

All published manuscripts reporting original research in Nature Portfolio journals must include a data availability statement. The data availability statement must make the conditions of access to the "minimum dataset" that are necessary to interpret, verify and extend the research in the article, transparent to readers

Authors must make available upon request, to editors and reviewers, any previously unreported custom computer code or algorithm used to generate results that are reported in the paper and central to its main claims. Any reason that would preclude the need for code or algorithm sharing will be evaluated by the editors who reserve the right to decline the paper if important code is unavailable.

Availability and peer review of computer code and algorithm

Authors must make available upon request, to editors and reviewers, any previously unreported custom computer code or algorithm used to generate results that are reported in the paper and central to its main claims. Any reason that would

preclude the need for code or algorithm sharing will be evaluated by the editors who reserve the right to decline the paper if important code is unavailable.

For all studies using custom code or mathematical algorithm that is deemed central to the conclusions, a statement must be included under the heading "Code availability", indicating whether and how the code or algorithm can be accessed, including any restrictions to access. Code availability statements should be provided as a separate section after the data availability statement but before the references

Version 1:

Reviewer comments:

Reviewer #2

(Remarks to the Author)

Thanks to the authors for taking the time to respond to all the points. I find the flow of ideas is much improved. And I appreciate the sketch of the system. My only comment at this stage is that the figures could perhaps be improved in places so that the fonts are all readable at sensible figure sizes.

Reviewer #3

(Remarks to the Author)

Reviewer #4

(Remarks to the Author)

In general, the comments raised in the previous version of the paper have been well addressed. This has greatly improved the quality and scientific rigor of the paper. From a scientific perspective, I only have a few additional minor comments to add, which can be found below.

Regarding data availability, I thank the authors for making an effort to share the code. A Google Drive link has also been provided with access to certain types of data. I would like to draw special attention to the high necessity of providing an appropriate and functional DOI before the paper's publication, once again to comply with the policy of Nature journals. Moreover, the repository only provides thickness change data, which represents only a small portion of the dataset used to conduct the scientific analysis of this study. One particularly critical piece of data is missing: basal melting. This is essential and must be included in the dataset. The absence of this specific and distributed cartographic information (in raster or NetCDF format) is critical, as it prevents users from replicating the analysis conducted in this study, hence verifying if the results can be reproduced.

If this is addressed (along with minor comments below), the paper will be ready published, and is a great contribution to our understanding of processing affecting ice shelf melting in Antarctica.

Minor comments.

Figure S2. You should still add the areas where basal melting was calculated.

Suggestion of figure: Providing a movie of changes in basal melting rates spatially through time would be great in illustrating the pulse in basal melting rates.

Fig 2e. What is the point of showing the evolution of PIG on this graph? The different evolution of the melting could also be related to the different geomorphological setup of the two glaciers (e.g. bathymetry...).

Fig S6. Could the lower melting observed at TEIS be explained by the separation of the part of the ice shelves by a ridge in the bathymetry, which could have prevented enhanced basal melting rates?

L339-340. Concerning the swath measurements, how much swath is used on the ice shelf? Don't you have phase ambiguity issues due to low surface slopes?

REVIEWER COMMENTS

Reviewer #1 (Remarks to the Author):

The paper is in good shape and describes important results that will be of interest to the ice-ocean interactions community as well as more broadly for interpreting and predicting future ice loss. The work is interdisciplinary, drawing from ice and ocean observations and from modeling results. I find the paper nearly ready for acceptance, and would support publication after minor revisions.

The key results show a novel event in which enhanced ocean melting of the Thwaites Ice Trunk is likely driven by an anomalously protracted and voluminous subglacial outflow event. The evidence from surface elevation data, when combined with ocean data and compared to modeling of the event is convincing that the elevated subglacial discharge is indeed responsible for the observations. The work also notes the evidence for presumed outbursts in the geological record as a contributing factor to past evolution of ice sheets, while paradoxically the results of this work suggest they do not likely determine the overall trends in ice sheet retreat. That connection is very important.

There are two general issues with the presentation of the paper; these are mostly superficial and thus under revision the paper would be much improved. First, there is little transition between sections of the effort--while using remote sensing and ocean data, there is little context given for the observations and their novelty within the main text, which should be improved. The methods are a place for detailed techniques but the main paper should describe better what the context of the observations are and how they are interpreted, not just delve straight into the interpretation. Similarly, the modeling shows up with not descriptions or context. Some lead in to the simulations describing them and their initial set up and purpose would improve reading of the paper.

Response:

We thank the reviewer for his review of our manuscript. Following the reviewer's comments we have revised the text to improve on the flow through the paper and on providing further context for the use of observation and modeling in support of our analysis. We respond more specifically to these points against specific comments below.

There are no significant flaws in the methods or interpretations. However, the last section of the paper trends toward speculation and could be improved with tightening of the logic and greater thought or direct commentary on whether any similar systems might exist within the observational record.

- Towards the model descriptions and context, they are somewhat brief even in the methods. Further detail on the modeling would improve the paper. Based on the written description, it is hard to know whether the models would be trivial to reproduce.

Response:

While the ice-sheet and ocean modeling approaches are brief relative to the other methods sections, this is because they are based on published approaches (Morlighem et al 2021 and Holland et al 2023, respectively) which have been modified incrementally in order to better suit the aims of our study. For example, in the ocean model the only change is to simply increase the subglacial discharge. The models are described fully in the aforementioned references, which would allow the results to be reproduced. We recognize however that more context and details need to be provided and have modified the paper and Methods accordingly. We also give better explanations of the context of the ice-sheet and ocean modeling within the main text of the manuscript.

Of minor but important note, several times in the paper a cavity that has recently been observed forming in the TWIT is referred to as the "Milillo Cavity" or similar; while it is a useful shorthand to describe features with names like this, especially in Antarctica where there is extreme sensitivity regarding how places are named, it would be better to find a different shorthand that is agnostic and not connected to a particular scientist. For example, the "new cavity" or "TWIT GL crescent" or something geometric or similar would be more appropriate (with all due respect to the work of Milillo et al who made an important discovery). Unofficial names with reference to individuals should be avoided in

general. Since the subglacial ocean environment proximal to the GL is already called a cavity, perhaps a new description would be more useful.

Response:

Agreed, in addition to the reference to a particular scientist, the use of “cavity” is also ambiguous in light of the use of the term to describe the location of seawater intrusions within the grounding zone. We replace “Milillo Cavity” by directly referring to the grounding line thinning and retreat at the TWIT grounding line embayment as was referred to by a few other authors.

I provided a commented document with tracked changes for other notes and comments on the main text of the paper, as well as a few comments below.

Response:

We thank the reviewer for detailed edits. We have accepted all suggested in-text edits when the text of the revised manuscript remains. We have reproduced, and responded to, the in-line comments of the commented document in our response below.

Minor comments:

Last sentence of abstract should be rewritten, as there is no evidence here or otherwise presented that would suggest that subglacial events would be stabilizing...so this sentence is challenging.

Response:

Here we use stabilising and destabilising in reference to the findings that subglacial flooding is less efficient overall at increasing ocean melting, i.e. a relative stabilisation compared to a trickling regime, and that subglacial flooding can trigger rapid grounding zone change. We propose to retain the reference to stabilisation and destabilisation but to more explicitly state what these terms refer to:

“As a result, it remains to be determined whether increased subglacial flooding events provide a stabilising influence on West Antarctic ice loss by decreasing the overall impact of subglacial water on ocean melting, or a destabilising influence by triggering rapid changes at the grounding zone.”

‘These’ in paragraph 1 (P1) is vague. Specify what is projected to increase, ie ‘While SLR contributions..’

Response:

Agreed, and changed as suggested.

Subglacial discharge section. Somewhere in here a statement of the observations would be appreciated. For example, just stating that surface elevations tracking hydrological flow were compared with ocean mooring data to confirm the influence of moving subglacial water.

Response:

We add the following introductory sentence: “Satellite derived ice sheet elevation change (see Methods) reveals that in early 2013 an extensive network of ...”

Beginning of P3, first sentence possibly needs a citation or several.

Response:

We removed this sentence as it repeated a point of the previous paragraph and the main point of the last sentence of P3. Citations supporting this statement are provided in the remainder of P3, and the last sentence in particular.

In the section on basal melt and ocean conditions, the description of surface elevation changes of TEIS and TWITgl at the end of the second paragraph is somewhat confusing. Looking at Figure S2, it is not clear that TWITgl recovers to where it would’ve been by 2018 and that TEIS doesn’t recover. See comments for Figure S2 for suggestions to increase clarity.

Response:

We modified the figure (now redesigned figure S6) as suggested by adding trend lines and amended the figure caption. The initial figure, with the trend lines, is shown below.

Third paragraph of the “Link between ...” section. Somewhere in here it is important to give a basic description of what was modeled. Presently, the text moves straight from observations to models with no pointer towards the how and why; it’s fine to cover most of the details but a transition into models and how the observations are coupled in is needed in the main text.

Response:

We now provide the following description: “To test the influence of subglacial lake drainage, we conducted a set of simulations of ice-ocean-subglacial interaction using the MITgcm ocean model (see Methods and ref. ³³ for a full model description). These simulations use an existing high-resolution model of the cavity beneath TWIT, which included subglacial discharge at the steady-state rate, assuming no flooding events³³. Here we use a Digital Elevation Model from 2013 for the TWIT geometry, and then run a set of steady-state simulations with different values of the subglacial flux, which is input at the deepest part of the Thwaites grounding line only (Figure S11d). These simulations suggest that ...”.

“Subglacial discharge can also have non-local impact via modification of ocean flow and stratification, and could account for the changes being observed further from the outflow region, under TEIS (Figure 1).”

-> Has this been shown to work? This would require an inversion of the flow

Response:

In Goldberg et al, 2023 (The Non-Local Impacts of Antarctic Subglacial Runoff) a regional model (~3 km resolution) is used to show that runoff from the TWIT grounding line impacts melt under nearby ice shelves (e.g. Dotson ice shelf, Fig 4 of that paper) by modifying stratification (Fig 6). The experiments in that paper did not examine the TEIS similarly due to the low model resolution but results suggest the mechanism is plausible. The high-resolution model in the present study does not include sea ice and cannot represent this nonlocal process. We agree that our statement regarding TEIS has not been shown definitively by the cited reference and therefore this is a speculative statement, and have tried to write it as such. We rephrase as follows:

“Subglacial discharge can also have a non-local impact on nearby ice shelves via modification of ocean flow and

stratification⁶⁰. We speculate that this process could account for the changes being observed further from the outflow region, under TEIS (Figure 1)."

Splitting this sentence into two and emphasizing that the second part is of a specific event that previously occurred would help with clarity. Defining 'moderate' with a specific value for how much melting occurred would also be helpful.

Response:

We rewrote this paragraph. The section of ocean condition now specifically quantifies the relative changes in heat content, via thermocline depth variation, providing context for the use of the term "moderate" in this later paragraph.

Are there any observed systems where this is either particularly likely, or where anomalous ice shelf change might be triggered this way? I am thinking about the Wilkns ice shelf for example.

Response:

We now cite the findings on Denman and Scott glaciers in Wilkes land (ref. 32: Pelle et al., 2023). We argue though that such potential conditions are ubiquitous across the Antarctic Ice Sheet. We revise the paragraph as follows:

"Such conditions are not limited to Thwaites Glacier, and are ubiquitous around the Antarctic margin, in particular at ice streams that drive the current Antarctic mass imbalance and will drive its future contribution to sea level³². Hence subglacial lake drainages, whilst infrequent, can likely play a role in modulating the ice response to ocean forcing, and therefore ought to be taken into account."

We also introduce examples from the Antarctic Peninsula and now of the Ross Ice Shelf in the second paragraph of the "Link between basal melt pulse, ..." section of the main manuscript.

Figures:

Figure 1a: Since each glacier is assigned one color it would make more sense to have a discrete colorbar rather than a continuous one. Continuous colorbar makes sense for panels b and c.

Response:

Agreed, we now provide a discrete colorbar for figure 1a.

Figure 3: Consider swapping this figure with Figure S4. Also could use a description of what the 'power law' is in the caption.

Response:

We would prefer to retain the figure 3 (now figure 4) as it illustrates the impact of the mode of subglacial water delivery on ice shelf melt rates. We added information about the power law in the caption as follows:

"Figure 4: Predicted additional basal melt rate due to subglacial discharge (modelled: red; power law approximation ($m = 3.78F^{0.54}$); subglacial flux multiple F is the subglacial flux used divided by $80 \text{ m}^3 \text{ s}^{-1}$, the background subglacial discharge rate for the main drainage network,..."

Figure S2: It would be helpful to show on the plot where the 2011-2013 rate of change falls so the readers are able to compare the observed values to something and to include a line on both plots showing where the elevation change would have been if following the 2011-2013 trend.

AND

Basal melt and ocean condition section: "Does this imply there is a lowering of the melt rates after 2014? Clarify/rework."

Response:

We updated figure S2 (now redesigned figure S6) as suggested, adding trend lines to both plots to better illustrate the point made about the impact of the transient melt event on the overall trend in ice shelf elevation.

With regards to “lower melt rates” yes, we modify the text as follows: “Over TWITgl a period of relatively lower thinning and melt rate takes place in the years following the melt pulse so that by 2018 surface elevation is at the level it would have been assuming constant 2011-2013 thinning rates (Figure S6).”

Figure S3 would benefit for some annotation showing readers where they should be looking for the polynya and how it is different/stands out. The outline doesn't match the outline in Figure 1

Response:

We updated figure S3 (now figure S8) circling the location of the polynya and updating the figure caption accordingly. The figure is produced from NASA worldview (<https://worldview.earthdata.nasa.gov/>) hence why the outlines differ from figure 1. We now also indicate the source of the images in the figure caption and mention in the data availability section the use of NASA worldview.

Figure S4 caption be consistent with /s and s-1, and with the timescale of the simulation.

Response:

Thank you, we fixed the units (now figure S11)

Figure S6 needs y-axis labels

Response:

Missing Y-axis fixed

REVIEWER COMMENTS

Reviewer #2 (Remarks to the Author):

The manuscript examines the Thwaites area and uses elevation changes, melt rate estimates and some hydrographic timeseries to consider interaction between the ocean, ice shelves, and a network of subglacial streams and lakes. The region is certainly critical for projections of future sea level.

I struggled as the manuscript does not clearly set up what it is going to examine and what the driving questions are. The introductory section just stops and then key scene setting points are made later on. The introduction is followed by subsections examining separate aspects of the data. Then finally a subsection comes along that reveals the possible connections between the various elements.

Response:

We thank the reviewer for raising this point and for suggesting ways to improve clarity and structure in the annotated manuscript provided along with your review. We have implemented your suggestions in the revised manuscript. The details on changes made to address clarity are provided below against specific suggestions.

What are the noteworthy results?

The key points seem to be - that there is interaction between the subglacial drainage and the shelf configuration and that none of this is particularly steady in time. To my mind the study doesn't establish the characteristic timescales for any of the processes. Instead, two months is described as quasi-simultaneous.

Response:

Here we show that the discharge of subglacial water triggers an increase in basal melt under Thwaites Ice Shelf and Ice Tongue. We show in the revised manuscript that subglacial flux and basal melt rates are strongly and significantly correlated and follow a power law relationship, a relationship that is commonly used to model the impact of subglacial discharge on ocean melting but that, to our knowledge, has never been observed under ice shelves. We show that melt rate increases from the time of discharge of subglacial water and that the peak melt rate follows the peak discharge by 2 months. The subglacial discharge takes place during a period of 8 to 10 months while elevated rates of basal melting remain for a few months after the end of the subglacial discharge. We show that the subglacial discharge and increased basal melt is very likely to have contributed to the retreat and acceleration of a sector of grounded ice situated at the grounding zone of Thwaites and highly sensitive to change in ocean melting and basal friction.

Will the work be of significance to the field and related fields? How does it compare to the established literature?

The region is certainly critical for projections of future sea level and there has been little work relating coastal processes to subglacial stream/lake transients that I'm aware of – I'm made a couple of reference suggestions.

Response:

Thank you for the suggestions, we've incorporated the references in the revised manuscript.

Does the work support the conclusions and claims, or is additional evidence needed?

The manuscript leaves the conclusions rather open-ended.

Response:

Again, thank you for raising this point and suggesting improvements. We have revised the text to try and strengthen the conclusion of our study

Are there any flaws in the data analysis, interpretation and conclusions?

I can't see flaws in analysis but as above the interpretation leave things open.

Is the methodology sound?

Yes

Is there enough detail provided in the methods for the work to be reproduced?

At a broad level probably

I think the story would benefit greatly from a clear sketch showing the system, the processes and the “direction of travel of effect”.

Response:

This is an excellent suggestion, this is now done as a new figure 5 added to the main manuscript.

There could be a lot more introduction and clarity when referring to various regions. More generally, I would have preferred more commas.

Response:

Thank you for the suggestions, following the detailed feedback in the annotated manuscript we have added commas and suggestions for improving clarity. The detailed response and actions to these comments are provided below.

There were no line or page numbers in my pdf so I marked the manuscript – unfortunately this seemed to result in a very large file size – apologies for that.

Response:

Thank you for providing an annotated manuscript. We have reproduced the comments below along with actions we have taken in response.

page and line numbers would be helpful

Response:

Yes indeed. Apologies for this oversight, the revised document now provides both line and page numbers.

Title:, Antarctica

Response:

Added

Abstract: isn't it more specifically to an _increase_ in ocean melting ... also ice shelf loss if 50% ice berg calving right?

Response:

This is a good point, we now mention “increase”. Iceberg calving is indeed roughly 50% of fresh water production. Here however we refer to the generic ocean forcing that can manifest itself by enhanced basal melting and ice shelf structural weakening.

Abstract: I think the surface elevation is about the only thing that is well constrained

Response:

We rephrased the abstract as follows:

“Upwelling from the release of subglacial melt water at the grounding line can enhance the ability of mCDW to melt ice shelves, but the efficacy of this process is still poorly understood and constrained and is currently not accounted for in projections of ice sheet loss.”

Abstract: is?

Yes

Abstract: at enhancing?

Yes

page 1: why capitals?

Response:

Good point, corrected.

Introduction, page 1: At the present time...

Response:

*We modified the sentence as follows: "The Amundsen ..., accounting for the vast majority of **the current** AIS mass loss^{1,5}"*

* Introduction, page 1: incursion to where?

Response:

*We rewrite as follows: "...enhanced incursion of modified Circumpolar Deep Water (mCDW) **onto the continental shelf, ...**"*

page 2: here're another couple that are likley relevant

Miles, B.W., Stokes, C.R. and Jamieson, S.S., 2018. Velocity increases at Cook Glacier, East Antarctica, linked to ice shelf loss and a subglacial flood event. *The Cryosphere*, 12(10), pp.3123-3136.

Whiteford, A., Horgan, H. J., Leong, W. J., & Forbes, M. (2022). Melting and refreezing in an ice shelf basal channel at the grounding line of the Kamb Ice Stream, West Antarctica. *Journal of Geophysical Research: Earth Surface*, 127, e2021JF006532. <https://doi.org/10.1029/2021JF006532>

Response:

Thank you for highlighting these two publications. We added the Whiteford publication in the indicated location in the manuscript. The publication on Cook Glacier addresses the impact of subglacial flood on grounded ice velocity, which we introduce in the discussion section relative to potential processes that lead to acceleration of the flow. We added the Cook Glacier reference then.

Introduction, page 2: right, and even if there were consensus it wouldnt be based on much evidence

Response:

Indeed, the text now explicitly mentions the lack of evidence.

*"..., there is **a scarcity of evidence and no consensus** on the efficacy of this process in impacting ice shelf melting, ..."*

page 2: at this point it would be normal to identify what question(s) the article is to examine/answer... what will we learn?

Response:

Agreed, we modified the introduction to add the following paragraph:

"Here we analyse change in ice elevation, ice velocity, ice shelf basal melt rate, and ocean conditions at the margin of Thwaites Glacier and within the wider Amundsen Sea sector during the discharge of subglacial water from an extensive network of lakes located under Thwaites Glacier in 2013. We pair these observations with modelling of the ice-ocean-subglacial system and of the sensitivity of grounded ice to ice-shelf and grounding-line change to assess the extent to which subglacial discharge impacts the rate of ocean melting and the stability of grounded ice at Thwaites Glacier."

page 2: vast is subjective/emotive scale - "extensive"?

Response:

Agreed, changed to "extensive".

page 2: for numbers less than 10 I was advised to spell them out

Response:

Agreed, and applied through the manuscript.

page 2: I'd take more time with Fig 1a to explain the network in spatial arrangement to the various ice shelves/glaciers etc

Response:

We have revised this section to improve the logical flow, which hopefully will help with the description of the spatial arrangement.

page 2: I suggest this could come into the main manuscript to strengthen the lake perspective.

Response:

Thank you for the suggestion. Since most of the lake timeseries have already been shown as part of previous publications, the cumulative lake timeseries is already shown in fig.2, and the individual lake behavior is not the main focus of the paper, we would prefer it to remain as figure S1.

page 2: I realise it is in the methods but a phrase to identify how this estimated e.g. "based on satellite-derived ice sheet elevation changes"...

Thank you for the suggestion, we added a sentence as suggested to indicate how this information was obtained:

"Satellite derived ice sheet elevation change (see Methods) reveals that in early 2013 ..."

page 2: reverse the structure of the sentence? "There was no evidence of elevation gain which would have indicated water storage downstream hence subglacial meltwater from the lakes presumably reached the grounding line downstream."

during the event as there is no evidence of elevation gain indicating water storage downstream

Agreed, changed as suggested.

page 2: over? from? through?

Changed "over" to "at".

page 2: you might structure this section in the same way as for SGD above - focus on the melt during the experimental period then contextualise it with the background conditions

Response:

Agreed. We rewrote this section to bring the focus on the experimental period, and integrating the context from the previously first paragraph where and when relevant.

page 2: glacier? or ice shelves?

Response:

"Ice shelves" would be ambiguous as there is a so-called ice shelf and an ice tongue, we modified to be more specific as follows:

"...melt rates under the Thwaites Eastern Ice Shelf (TEIS) and near the grounding line of Thwaites Western Ice Tongue (TWITg) ..."

page 2: than what?

We reworded this section.

page 2: damage? in what sense?

and

reword/clarify

Response:

This is now removed after the section re-organisation and instead the information is provided, with clarifications, in the method section

page 2: "sector" gets used lot - I can see how it works for Amundsen Sea Sector (equivalent to region) - but for all these other sectors - how are they defined and separated?

Response:

Agreed, in this instance "sector" is unnecessary, we simplified it to "Thwaites". In response to a reviewer's comment we simplified the breakdown of sectors into a more conventional one i.e. individual ice shelves and, for Thwaites, its Eastern Ice Shelf and Western Ice Tongue.

page 3: Basal melt is typically driven by...

Response:

Agreed, changed to suggested wording

page 3: this sentence would seem well suited to a paragraph above at the end of the Intro where you define that actual problem and the questions...

Response:

We now provide a final paragraph in the introduction where the problem, question, and methods, are introduced.

page 3: here should you connect to melt/ablation parametrisation? ... and tides might be considered as a process that contributes to melt

Response:

With respect to existing melt rate parametrisation using ocean heat content we added a reference to Hill et al., 2024. Whilst tidal energy is generally an important contributor to melt, it varies significantly from ice shelf to ice shelf, it is expected that tides play a relatively minor role under Thwaites (Jourdain et al 2019, 10.1016/j.ocemod.2018.11.001) when it comes to ice shelf melt. Recent work noted the potential importance of under-ice intrusions of ocean waters

on a broad grounding zone at the tidal frequency, we have added a paragraph in the last section (“Impact ...”) of the revised manuscript. Our observations do not allow us to shed light on these processes.

page 3: hydrographic moorings

Agreed, changed to suggested wording

page 3: any of the related studies show how relevant this location is for the Thwaites region?

Response:

Wahlin et al 2021 discusses circulation within the Amundsen Sea. We reinforce this aspect by adding two additional references by Nakayama et al 2019, and Dotto et al 2022, and by being more explicit in the text.

“Basal melting is typically thought to be driven by ocean heat content (OHC) and circulation under ice shelves^{41–43}. Two hydrographic moorings located in front of the Pine Island ice shelf, and indicative of oceanic variation impacting both Pine Island and Thwaites ice shelf cavities^{44–46}, show that the thermocline depth, a proxy for ocean heat content, deepens from 600 m to 750 m between 2011 and 2013 (Figure S7), and oscillates around a depth of 700 m during the following five years (Figure 2e, Figure S7).”

page 3: is it a good proxy for OHC if it is missing upper ocean?

Response:

The upper ocean has limited to no access to the cavity. This is especially true for the deeper parts of the cavity, which should mainly be concerned with heat content >400m depth. Whilst moored observations do miss the upper 350m of the water column, the thermocline separating surface and bottom water can clearly be identified at all times. The OHC recorded by the mooring is therefore a good proxy for the glacial melt available OHC outside of the cavity.

We have added the following text to the methods section relative to the ocean observation:

“While the moorings are missing the upper ocean, the upper ocean has limited to no access to the cavity. This is especially true for the deeper parts of the cavity, which should mainly be concerned with heat content >400 m depth. Whilst moored observations do miss the upper 350 m of the water column, the thermocline separating surface and bottom water can clearly be identified at all times. The OHC recorded by the mooring is therefore a good proxy for the glacial melt available OHC outside of the cavity.”

page 3: deepens

Agreed, changed to suggested wording

page 3: clarify???

This section has been completely rewritten following the correlation analysis performed during revision.

page 3: ??? unclear sentence structure

We rephrase as follows:

“In early September 2013, a polynya started forming about 20 km offshore from Thwaites’ grounding line (Figure 1.a and Figure S8), reaching a maximum extent of 86 km² by November 2013.”

page 3: very small as far as polynyas go

Yes, but typical for a sensible heat polynyas formed by upwelling plumes, e.g. at the front of Pine Island Glacier in the references cited. We now make a mention of “sensible-heat” in the manuscript to distinguish from latent-heat polynyas.

page 3: via what mechanism? AND why?

We rephrase this section to make the inferences more explicit:

“The polynya is embedded within undisturbed sea ice and icebergs and as such differs in both size, aspect, and timing from large wind-driven polynyas frequently occurring in the sector, but resembles sensible-heat polynyas associated with subsurface ice-shelf outflows, which bring warm deep waters to the surface^{48,49}. This polynya is unique in the recent observational record at Thwaites Glacier (Figure S8) and develops during the peak of subglacial discharge, slightly preceding the peak in basal melt rate increase under Thwaites. Thus we speculate that the polynya is the consequence of the buoyant plume created by the subglacial discharge and meltwater, entraining deep ocean heat, delivering it to the ocean surface, and melting the sea-ice.”

page 3: pedantic but does the "grounding line" thin?

We rephrase as follows:

“Rapid grounded ice thinning and grounding line retreat”

page 3: suggest re-writing the entire sentence - maybe split into two

We rephrase as follows:

“Between 2014 and 2017, the western side of an embayment in the TWIT grounding line experienced a combination of strong localised grounded ice thinning, flow acceleration, and grounding line retreat^{38,50,51} (Figure S9).”

page 3: re-order words for clarity

Addressed as part of the response to the previous comment.

page 3: decreased ... was reduced... ?

Replaced by “surface lowering”

page 3: ,

Added

page 3: again a phrase to indicate how this was derived.

We added the following in the previous paragraph when elevation change is first introduced:

“Our observations of satellite-derived ice sheet elevation changes (Figure 2c) show that, prior to thinning and retreat, the area was lowering at an average of 4.4 m yr⁻¹, consistent with the general rates of dynamic thinning reported at the ice sheet margins^{52,53}. From approximately April 2013 the thinning accelerated, with the height of the ice suddenly lowering at rates of 11.7 m yr⁻¹. From 2017 onward, rates of surface lowering dropped to 1.2 m yr⁻¹ indicating that the area is now largely ungrounded.”

page 3: isn't it the melting that got us the newly exposed ice via retreat?

Yes this is correct, but in this section we refer to the geometrical feedback described in Holland et al., 2023 (ref. 32) during the period covered by our study. To avoid ambiguity, we make an explicit mention of the term used in Holland et al:

"We note that changes in geometry led to enhanced basal melt under Thwaites via "geometrical feedback"³³, particularly due to the onset of melting under the areas newly exposed to the ocean and due to changes in ice basal slope."

page 3: is this an ice shelf or what? And is it the only retreat observed?

As suggested by reviewer#1, we replaced the reference to the "Milillo cavity". The sentence now reads:

"This melt increase took place progressively from 2011 to 2017, with a higher melt increase between early 2014 and late 2015 reflecting the rate of grounding line migration along the western side of the grounding line embayment at TWIT³³."

Retreat took place elsewhere along the grounding line of Thwaites, but the unpinning is specific to this sector.

page 3: So far we've had to guess that this was where we were going to arrive

The last section of the introduction now hopefully provides a useful early indication of the direction of travel.

page 4: what is the characteristic timescale for the subglacial hydrology - a few months seems quite a long time to me for a hydrostatic process

Agreed. The mention of a "few months" was more a reflection of the duration of the events, i.e. lake discharge, grounding line thinning and retreat, transient increase in ice shelf melting take months to develop and subside, as well as the uncertainty in the relative timing of events from the satellite observations constraint, but we agree that it is unhelpful and ambiguous as a description of the temporal relation between events and hence modified the sentence.

From the lake drainage we can gather that subglacial outflow took place uninterrupted for a period of about a year, with about 6 months between the first signs of lake drainage in early 2013 and the peak discharge in August-September 2013. Ice thinning, ice acceleration, and basal melt rate increase, all started at the onset of lake drainage. We believe that figure 2 gives a good visual aid to appreciate the relative timing and magnitude of events. We further added the subglacial flux curve to figure 2e to illustrate the relative timing of subglacial lake discharge and basal melt. We also added a correlation and lag analysis between lake discharge and basal melt rate to the study, that provides further indication of the relative timing of the 2 events. This is shown in the revised manuscript as a new figure 3, and a new paragraph in the section "Link between basal melt pulse, ...":

"We observe a significant correlation between the timeseries of subglacial flux and the basal melting under Thwaites, driven by the 2013 changes (Figure 3a). Correlation between subglacial flux and basal melting is increased even further by considering a ~3-month time-lag (Figure S10 and Figure 3b). The nature of the 3-month lag is unclear but appears to be related to the differing length of the pulse in subglacial flux and basal melting, while the start of basal melt increase and subglacial flux release appear to be synchronous, the peak in basal melting takes place about two months after the peak in subglacial flux, and basal melting takes longer to return to its baseline value following the end of lake activity."

page 4: We have to wait to here to look at the first panel of Fig 2 - which to me is where all the pieces are laid out yet it is mentioned here almost in passing... There's a reason why it sits at the top of this figure and so I'd start with that.

The manuscript is structured so that each event is first described individually, with the penultimate section discussing the connection between events. This section starts with figure 2a, which as you say, lays all the pieces out. This feels logical to us, and hopefully with the new paragraph ending the introduction section the penultimate section on connectivity is now less of a surprise.

page 4: a phrase to explain?

We develop as follows:

"It is plausible that grounding line thinning and retreat, by locally changing the ice surface, hence the hydraulic potential, and shortening the grounded ice between the lakes and the grounding line, would lead to a change within the subglacial system and act as a trigger for lake discharge⁵⁴⁻⁵⁶."

page 4: this seems the critical question for the whole article - can this be expanded upon significantly - in terms of clarification?

We have revised and expanded this section to improve clarity as follows:

"It is plausible that grounding line thinning and retreat, by locally changing the ice surface, hence the hydraulic potential, and shortening the grounded ice between the lakes and the grounding line, would lead to a change within the subglacial system and act as a trigger for lake discharge⁵⁴⁻⁵⁶. If this were the case, we might expect lakes nearer to the grounding line to be impacted first, with lake activity progressively migrating upstream. Although there is uncertainty in the relative timing of drainage^{36,39} all evidence points instead towards a cascading drainage with the most upstream lakes discharging first and initiating a cascade of lake drainage propagating downstream³⁹. We also note that the two most upstream lakes Thw170 and Thw142 experienced a second episode of drainage in 2017 with no apparent grounding line change and with the lower part of the subglacial system seemingly shut down, suggesting that the subglacial system under Thwaites can be activated by triggering mechanisms unrelated to grounding-line change. Furthermore, most of the rapid grounding line retreat and thinning takes place further downstream of the modelled location of subglacial outflow through the grounding line (Figure 1b), hence having a relatively small direct impact on the hypopotential over the drainage network itself."

page 4: where is this sector and what defines it?

This referred to the Milillo cavity, however this has been rephrased following a suggestion by another reviewer.

page 4: I thought "lake discharge" WAS the event we were looking for?

We agree that this sentence is confusing, we clarify as follows:

"Lake discharge is also a potential trigger for the changes observed at the grounding line and under TWITgl and TEIS."

page 4: Which?

Changed to: "Implication for the stability of Thwaites Glacier"

page 4: seems like a good point to make in the Intro

Agreed, we now include mention of sensitivity in the wider context of Amundsen Sea Sector change, and in the final paragraph of the introduction.

page 4: using what? of what?

Agreed. We changed to:

"To test the influence of subglacial lake drainage, we conducted a set of simulations of ice-ocean-subglacial interaction using the MITgcm ocean model (see Methods and ref. ³³ for a full model description). These simulations use an existing high-resolution model of the cavity beneath TWIT, which included subglacial discharge at the steady-state rate, assuming no flooding events³³. Here we use a Digital Elevation Model from 2013 for the TWIT geometry, and then run a set of steady-state simulations with different values of the subglacial flux, which is input at the deepest part of the Thwaites grounding line only (Figure S11d). These simulations suggest that the 2013 lake discharge could have caused ..."

page 5: the structure seems reversed? I would have thought we'd consider SGD early on and then come to downstream impacts?

Agreed, we now invert this section and the one immediately above discussing downstream impact.

page 5: what constitutes a "pulse" - this seems quite long based on my pre-conceptions from other systems.

We use the term to describe the relatively short-lived nature of the melt increase, in contrast with for example a more progressive transition between cooler and warmer ocean regimes, taking place and sustained over multiple years in the Amundsen sector. As discussed above the proposed source for the melt increase, the subglacial water outflow, is taking place over a period of several months.

page 5: ,

Added

page 5: ,

Added

page 5: oceanic plumes?

"plumes" Added

page 5: I've not come away with any enhanced understanding of this frequency...

True, although in this study we do not address the frequency of lake drainage other than with the simulation in effect testing the impact of lake discharge taking place every year to every 8 years. There are several publications looking at the frequency of lake discharge in Antarctica in the satellite and geological record and this is addressed in the second paragraph of the last section of the manuscript.

page 5: a subheading for SGD flow rate estimation?

Yes, we added a section to the method section to describe the subglacial modeling.

page 6: how does this compare with available data?

We have now expanded the description of the model in the Methods section, and also noted that a comparison to the available data is discussed in the original paper (ref. 33).

page 7: oC?

Replaced with suggested notation

page 7: where to we learn about the two different lengths?

We now provide an additional supplementary figure, figure S7, with the two moorings' records.

page 7: suggest the subheading refer to the methods not the science

Agreed, we replace with "Ice sheet modeling"

Fig 1a: I know its common knowledge but there is no indication of where this is or scale... no lat/lon, no inset map of the continent etc

An inset map was added.

Fig 1, caption: a bit more guidance as to where.. and what kind of mooring?

Thank you. We added details about moorings location and type. Additional details are also provided in the method section.

Fig1: b and c would seem supplementary type material

We would prefer to retain them in the main manuscript as they are essential to the argument.

Fig2e: Its not "moorings" is an isotherm depth

AND

Fig2e: move "Thwaites melt" label so that it is not right by the moorings line

AND

Fig2a: where is the subglacial network?

AND

Fig2a: where are the lakes in all this?

Thank you. We modified and clarified labels and location of various references in figure 1 and 2. The lakes are implicit in the subglacial network at the proximity of the grounding line, it would be difficult to represent all lakes here and we feel this information is conveyed between figure 1 and figure 2.

page 12: nothing in caption refers to the two curves ... why have 2 if they are identical? Error/variability bars?

Good point, the main text and method refers and provides context to the power law approximation but not the caption! The caption now explicitly refers to the 2 curves and provides the values of the power law relationship. The performance of the model results is discussed in ref. 33 and is now also mentioned in the corresponding methods section.

REVIEWER COMMENTS

Reviewer #4 (Remarks to the Author):

The paper by Gourmelen et al. describes a lake drainage event that occurred on the Thwaites Glacier in the Amundsen sector of West Antarctica. Although the methodological developments of the paper are not groundbreaking and are rather standard, this study highlights a unique event that could potentially impact the stability of the Thwaites Glacier, if it becomes more frequent in the future.

Overall, the paper is well structured and well written. Lines and page numbers should be added to facilitate the review process, right now it is really difficult to reference correctly the text in the review. There is still work to be done on the figures, which sometimes do not provide the necessary information to support the arguments made in the text (see comments below). Critical methodological information is also lacking at the moment (see specific comments below). Finally, the paper should better discuss recent findings by Rignot et al., 2024; which have highlighted significant grounding line modulation in this region that could challenge the proposed processes of this paper.

More importantly, the paper does not satisfy the open-science requirement of nature journals, and nature communications more specifically. This is currently a really concerning issue. There is not even a data availability/code sharing statement. There is no sharing of any of the results, and no sharing of any of the codes to retrieve the data. It is therefore not possible to verify or reproduce any of the results that are presented here, which is a major obstacle for the paper review.

As a consequence, I would formally oppose the publication of this study, until the data necessary to verify the methodology and results have not been provided. This includes: - maps of basal melting rates throughout the entire study period, elevation maps both from Cryosat-2 swath data or POCA, and codes to replicate the processing performed by the authors (swath processing, retracking method), along with CSVs for all data used in the figures.

Response:

We thank the reviewer for taking the time to thoroughly review our manuscript. The general points the reviewer raises above are further detailed in the specific comments below, hence we provide our answers to them specifically as they are raised. Regarding the data and code availability, we apologize for this oversight and have now added a code and data availability repository and related sections in the manuscript. We provide further responses to this point against specific comments below.

Specific comments

- The paper needs a small section on the methodological approach and the overall objective of the paper before running into the result section.

Response:

Thank you for raising this point, in response to yours and similar comments by the other reviewers, we have added a closing paragraph to the introduction section:

"Here we analyse change in ice elevation, ice velocity, ice shelf basal melt rate, and ocean conditions at the margin of Thwaites Glacier and within the wider Amundsen Sea sector during the discharge of subglacial water from an extensive network of lakes located under Thwaites Glacier in 2013. We pair these observations with modelling of the ice-ocean-subglacial system and of the sensitivity of grounded ice to ice-shelf and grounding-line change to assess the extent to which subglacial discharge impacts the rate of ocean melting and the stability of grounded ice at Thwaites Glacier."

- Add error bars to the changes in elevations (Fig S1), this should also include penetration uncertainties

Response:

Good point, we added uncertainty shading to Figure S1 as suggested.

- Please precise how you calculated the channelized system

Response:

We added a dedicated section "Subglacial routing" in "methods"

- How did you derived the changes in surface elevation ? Over which area ? Is this an average ? Single point ? Please add on figure 1 the area where you calculated the trends.

Response:

Individual lake volume change (Figure S1) is determined by integrating the elevation change over the area of each lake displayed in Figure 1. The total volume change is then the sum of all seven volume change timeseries. This is detailed in refs. 37 and 39 from which these observations are extracted. We have added a section in the methods as follows:

"Volume change over the subglacial lakes (Figure S1) is derived by integrating the elevation change over each of the seven lakes area displayed in Figure 1a^{37,39}. Figure 2d shows the volume change sum of all seven lakes. The change in elevation over ice shelves displayed in Figure 2 corresponds to the mean elevation for the areas shown in Figure 1 and Figure 2, values for "Thwaites" corresponds to the sector including both TEIS and TWITgl."

All areas used for analysis (i.e. lakes, ice shelves, grounding zone retreat) were already shown in figure 1, which is now stated more explicitly in the methods section and captions. We also provide the masks in the manuscript's dataset.

- How often do you have measurements of surface elevation, ie how confident are you with the timing of lake drainage? Please be more specific.

Response:

We have now added an additional figure as supplementary material providing the monthly data density over each of the lakes (Figure S12). We added the following sentence in the "elevation and elevation change" methods section:

"CryoSat's orbit and swath processing help to retrieve elevation at monthly intervals even over relatively small targets such as sub-glacial lakes (figure S12)⁷³."

- Can you estimate radar penetration biases using ICESat-2 for confidence over 2018-present ?

Response:

The reviewer is right to raise this point as it is a key question in microwave-based studies of the cryosphere. We are fully aware of this, hence why we provided, in the initially submitted manuscript (supplementary material, section 7c, now section 14), an argument why the observed "pulse" of change over the floating Thwaites cannot be caused by change in the depth of the scattering phase center; it is essentially too large and of the opposite direction to what is expected due to anomalous melt events. This is also why we use a well-known approach, the TFMRA retracker, to minimize the impact of fluctuating penetration.

Fluctuation in radar penetration can however affect the uncertainty characterisation. Penetration fluctuations are already partly accounted for in our monthly height uncertainty by making use of the residual data spread. Co-analysis with laser altimeters can provide a way to assess the degree to which fluctuation in scattering depth can impact the accuracy of the measure of elevation change. In fact there have been numerous studies already analysing differences between Ku-band altimeters and either Operation Ice Bridge or ICESat-2 in the context of swath and TFMRA retracking¹⁻⁸. These require careful consideration to extract the penetration component from all other potential sources of difference between the radar and the laser that have to do with either other sources of error, or that can stem from real differences related to e.g. differences in spatial and temporal characteristics of the missions.

In the revised manuscript, and following the reviewer's suggestion, we apply an additional uncertainty term to the elevation dataset. This is described in the revised methods section, and accounted for in the revised figures. To our knowledge, the most comprehensive study of transient penetration bias due to change in firm properties, at the monthly time resolution, is that recently performed by Helm et al., 2024. As a conservative approach we use their high-end value of 0.3 m error and add it in quadrature to the other uncertainty terms of our monthly elevation change estimates. This term impacts the uncertainty of elevation change and basal melt rate. The impact is minimal for Thwaites which had already relatively large elevation uncertainty.

We also compare our melt results with a recently published high-resolution melt study of Thwaites based on REMA DEMs (ref. 40 i.e. Chartrand et al., 2024) and found close agreement. We added the following to the main text and Methods section:

Main:

"Our melt rates are in close agreement with a recent high-resolution study⁴⁰ (see Methods)."

Methods:

"Our melt rates agree well with a recent high-resolution melt study⁴⁰, mean basal melt rate of overlapping measurements is 20.4 m yr⁻¹ for this study, and 21.8m yr⁻¹ and 21.6 m yr⁻¹ for ref.⁴⁰'s 2011-2015 and 2016-2019 periods respectively. Ref. ⁴⁰ also found relatively higher rate of mean basal melting under TWITgl during a period including the basal melt pulse discussed in this study i.e. 66.9 m yr⁻¹ (2011-2015) compared with 52.5 m yr⁻¹ (2016-2019) and 55.7 m yr⁻¹ (2020-2023)."

1. Nilsson, J. et al. Greenland 2012 melt event effects on CryoSat-2 radar altimetry. *Geophys. Res. Lett.* **42**, 3919–3926 (2015).
2. Nilsson, J., Gardner, A., Sandberg Sørensen, L. & Forsberg, R. Improved retrieval of land ice topography from CryoSat-2 data and its impact for volume-change estimation of the Greenland Ice Sheet. *The Cryosphere* **10**, 2953–2969 (2016).
3. Gourmelen, N. et al. CryoSat-2 swath interferometric altimetry for mapping ice elevation and elevation change. *Adv. Space Res.* **62**, 1226–1242 (2018).
4. McMillan, M. et al. Increased ice losses from Antarctica detected by CryoSat-2. *Geophys. Res. Lett.* **41**, 3899–3905 (2014).
5. Morris, A., Moholdt, G., Gray, L., Schuler, T. V. & Eiken, T. CryoSat-2 interferometric mode calibration and validation: A case study from the Austfonna ice cap, Svalbard. *Remote Sens. Environ.* **269**, 112805 (2022).
6. Helm, V., Humbert, A. & Miller, H. Elevation and elevation change of Greenland and Antarctica derived from CryoSat-2. *The Cryosphere* **8**, 1539–1559 (2014).
7. Gray, L. et al. CryoSat-2 delivers monthly and inter-annual surface elevation change for Arctic ice caps. *The Cryosphere* **9**, 1895–1913 (2015).
8. Helm, V. et al. AWI-ICENet1: a convolutional neural network retracker for ice altimetry. *The Cryosphere* **18**, 3933–3970 (2024).

- Please be more specific of the routine modeling. How was this calculated ? You cannot expect the readers to know everything, you have to provide a certain level of details.

Response:

We added a dedicated section "Subglacial routing" in the methods section.

- "During this event, all water...network." Which lake are you referring? Can you see an elevation gain which would justify that everything has been collected there? Please provide more details.

Response:

We now indicate that lake drainage takes place at the two most upstream lakes of the main subglacial network, Thw170 and Thw142. We also provide, in that sentence, the reference where this is analyzed in more detail. We finally refer to the figure in the supplementary material where the elevation gain is displayed.

"A second episode of lake discharge took place in 2017 at the two most upstream lakes of the main branch of the subglacial network, Thw170 and Thw142³⁹. During this event, there is surface inflation above the lake immediately downstream, i.e. Thw124, coupled with no noticeable activity further downstream in the network (Figure S1)³⁹."

- "Satellite-derived oceanic melt rates... 10.8 m/yr." Over which area did you calculate this average melting rate? Since a large part of Thwaites has collapsed it is of prime importance to understand where you calculate this change in melting rates.

Response:

Yes this is of prime importance as Thwaites is rapidly disintegrating and so the extent over which the analysis is performed matters a great deal. This is why we have excluded from the analysis the areas of Thwaites that collapsed or retreated during the study period, hence why TWITgl, over which we perform the analysis, is significantly smaller than the wider TWIT usually referred to in the literature.

Figure 1 shows the area over which melt rates have been calculated. The methods section (copied below) provides further information. We've amended the text and figure caption, in particular figure 2e, to make sure that the numbers are more clearly linked to the areas considered. We also provide the masks of the areas used in the manuscript dataset.

"We exclude from the calculation areas where a high shear rate indicates the presence of fractured ice that would potentially bias the surface elevation change and where the mass conservation approach breaks down⁸³. This means that a large proportion of TWIT, beyond the TWITgl sector, is excluded from this analysis. Significant episodes of grounding line retreat have taken place in the Amundsen Sea region since 2010. To ensure that we only consider elevation measurements made over floating ice for the duration of the CryoSat-2 period, we use the 2011 grounding line definition¹⁰. Similarly, we consider the minimum ice shelf extent during the 2010-2021 period to ensure that melt variability is not affected by difference in spatial extent of ice shelves. However, this means that we do not capture melt in area that have ceased to be ice shelves, or that have become ice shelves, between 2010 and 2021⁷⁴. The resultant mask means that changes in elevation and basal melt are reflective of the processes affecting ice that is in contact with the ocean for the duration of the observed period."

- "near the grounding line (Figure 1a)". Are you referring to Figure 1b?

Response:

The reviewer is correct. This section has been restructured during the revision, and this is no longer mentioned in the text.

- "TWITgl is characterized by a deeper... average of TEIS." Please provide a figure to justify that statement.

Response:

We now provide a figure S4 displaying the draft and a correlation analysis of basal melt rates versus draft, and specifically refer to the figure at this instance in the main manuscript.

- "Basal melt rates under Thwaites...double (Figure 2e)". Please also provide the basal melt rate time series instead of the anomaly as a supplementary figure. Right now it is difficult to see if it doubled or not. Figure 2e only show the anomaly for what is called "thwaites" in general. So it is a bit unclear to see at what location the time series has been calculated. Please provide it also for the different subregions, in order to look at if we can observe the pulse everywhere.

Response:

As suggested we provide the basal melt rate timeseries as a supplementary figure, figure S2. We also now clarify, in the basal melt rate section of the manuscript, as well as in the method section, the area over which the quantities are calculated, described in more detail in a response to a previous related comment on elevation change. Figure S3a and figure 3b provides timeseries of basal melt and ice velocity highlighting the change in basal melt and surface elevation during the event for both TWITgl and TEIS. Note also that in the original manuscript the melt axis of figure

2e was mislabelled, which also impacted the related values in the text, this is corrected in the revised version of the manuscript. We also improve on the basal melt calculation over TWITgl, from the original dataset we generated as part of the Davison et al., 2023 study. In Davison et al., 2023 we used the ITS_LIVE multiyear composite to derive ice divergence, this dataset records the disintegration of the TWITgl sector in later years, this in the divergence field leads to apparent regions of refreezing hence the impossibility to determine melt rate at these locations and the absence of melt observation in the Davison et al dataset. In the revised version of our manuscript we use a 2015 velocity field (see method) that is free of this issue and allows basal melt determination over most of the TWITgl sector (new Figure 1b). The method section, main text, and figures have been updated accordingly.

- "The pulse in basal melt rate is observed...TEIS and TWITgl". I am a bit sceptic about the TEISgl and TEISfront zones. What criteria was used to differentiate both regions?

Response:

Fair point, we thought long and hard about this. We chose these sectors to investigate and illustrate the combined impact of distance from source + ice draft. Since there is little significant difference between TEISgl and TEISfront, in the revised version we decided to merge these two zones into just one TEIS (figure 1 & 2). However, as it may be of interest, we still provide a figure with the split TEIS front and TEISgl as supplementary material (figure S5).

- "During the melt pulse event...(Figure 2b)." It is really difficult from the provided figures, to assess that these statements are convincing. 1) Figure 2 caption says it is cumulative elevation change anomalies, but the legend only says elevation anomalie, and the text describing figure 2b talks about elevation. Please make things clearer.

Response:

We modify this section for clarity, and we also add a line to the methods section to describe what is meant by elevation anomaly. We note that in the quoted text from the manuscript "elevation" is used to describe Figure S3, that indeed shows elevation change and not anomaly.

In the caption of figure 2b we now refer to elevation change anomaly, as in the legend.

In the text, we now write: "During the melt pulse event, Thwaites ice surface lowers rapidly by 3 m on average over TWITgl, and 1 m over TEIS (Figure 2b, Figure S5). TEIS does not appear to recover after the transient melt event where surface elevation remains lower than it would have been based on the 2011-2013 rates of change (Figure S6). Over TWITgl a period of relatively lower thinning and melt rate takes place in the years following the melt pulse so that by 2018 surface elevation is at the level it would have been assuming constant 2011-2013 thinning rates (Figure S6)."

In the methods section we now write: "Elevation change anomaly correspond to the detrended change in elevation and are shown as elevation anomaly from the first 2 years of observation."

- "28 m on average over the TWITgl". I don't really see how this is being retrieved from figure 2b. At the end of the melt pulse event, I can see that the elevation change anomaly over TWITgl is -3 meters. How do you calculate these anomalies? With respect to what year? Please provide further details.

Response:

We agree with the reviewer that this mention of thickness change is confusing. The text now refers to the change in elevation, shown in the figure and measured, rather than thickness change. As noted above we have added more details in the method section about how the anomaly is calculated.

- "TEIS does not appear to recover... rates of change (Figure S2)". From Figure 2e, the elevation change anomalie remains around zero for TEIS, so I don't really understand the fact that TEIS does not recover.

Response:

The reviewer is correct, figure 2e is the anomaly from the trend across the entire observation period. To make the point clearer, and as suggested by another reviewer, we added trend lines to Figure S2 (now redesigned figure S6) to show that elevation change at the 2010-2013 rate would lead to a higher elevation than observed for TEIS at the end of the observation period, and the same elevation than observed for TWITgl.

- "Over TWITgl...thinning rates (Figure S2)." Did the author switched TWIgl and TEIS ? because based on this description this is not what Fig S2 is showing at all.

Response:

We have now added trend lines to figure S2 (now figure S6) which should make this statement clearer.

- "thermocline depth...in the following 5 years". The thermocline depth increased by how much?

Response:

We rephrased this section as follows:

"Two hydrographic moorings located in front of the Pine Island ice shelf, and indicative of oceanic variation impacting both Pine Island and Thwaites ice shelf cavities⁴⁴⁻⁴⁶, show that the thermocline depth, a proxy for ocean heat content, deepens from 600 m to 750 m between 2011 and 2013 (Figure S7), and oscillates around a depth of 700 m during the following five years (Figure 2e, Figure S7)."

- Figure 2e. The dotted lines are the basal melting rates, but the Thwaites melt label is placed right on the moorings. Please have a proper legend. Plus it is impossible to really see if PIG melt follow the thermocline depth. Indeed the basal melt anomaly is close to zero most of the time. We have a slightly positive anomaly in 2013, but after that, anomalies seems always negative while the thermocline depth continued to increase. After 2016, the thermocline depth decreased but it seems the only year when the PIG melt anomaly looks positive again. A larger graph on this specific periods is needed to conclude.

Response:

We significantly revised the section about the relationship between thermocline depth and basal melt timeseries. As part of this revision we performed a correlation analysis (new figure 3) that indeed shows no significant correlation between thermocline depth and basal melt, both for PIG and Thwaites glaciers. It is fair to say that (1) the relationship between ocean heat content available to melt ice shelves and melt itself is not obvious, as processes such as geometric feedbacks and ocean circulation changes can modify this relationship, and (2) satellite derived basal melt estimates come with uncertainties. Here we show the complexity of the relationship, the text of the main manuscript now reflects this also.

We added a discussion paragraph to the main text highlighting some key points:

"We found no significant correlation between the thermocline depth and basal melt rates under Thwaites and Pine Island across the entire study period (Figure 3a). This lack of correlation suggests that the relationship between melting and OHC is not straightforward, or is not well represented between our moorings and satellite melt derivation (Methods). Geometric feedback and ocean circulation, on the continental shelf but also at the ice front and inside the cavity, or strong ocean stratification, can modify this relationship^{31,33,47}."

We also revised figure 2e to improve visibility, creating a distinct legend section as suggested.

- "Similarly, basal melt rate under Thwaites...". This is not obvious since the basal melt anomalies are negative or close to zero from Fig 2e

Response:

Based on the results described in the previous response item, this section is now rewritten. We also now provide a figure with the melt rate, and not anomaly, as figure S2

- "10 km by 5 km". Please provide an area of the polynia instead. This does not mean anything since the polynia is not a square.

Response:

The polynya reaches a maximum extent of 86 km²; this is now added to the text.

- "Between 2014-2017". If you talk about the Milillo cavity as referred in the figure, please state the reference name here.

Response:

Based on another reviewer's request we now refer to this area in a different way, stating the area of 2013 grounding line retreat and thinning. The text and figures have been amended to reflect this.

- "During this event, surface elevation...(Figure 2c)". This is not what Figure 2c is showing, please also provide time series of basal melting rate on top of anomalies.

Response:

The reviewer is correct, this first paragraph mixes results from previous studies, which we reference, and that of ours, which is confusing. We simplify the structure of this section by restricting the first paragraph to what was known prior to this study, and focus the second paragraph on the new information gathered.

- "4.4 m/yr". Please indicate the rates close to each of the fits that you show in Fig 2c.

Response:

Both rates are now added to the figure.

- "While ocean warming has been advocated...". The lake volume change, shown in Fig 2d, is really coherent with basal melting increases and the ocean temperature increases. The exact timing of the event has an errorbar (return period of cryosat), which has to be considered and might explain why the peaks does not exactly corresponds.

Response:

We have provided further analysis on this matter. CryoSat-2 has 30 days sub-cycles, hence providing monthly observation - a new supplementary figure S12 illustrates the monthly coverage over even relatively small spatial extent like over subglacial lakes, this is also now mentioned in the methods section. We also performed correlation analysis of the melt, subglacial flux, and thermocline which shows no significant correlation between thermocline depth and basal melt. A more detailed description of the correlation analysis, and the changes made to the manuscript, is presented in response to a comment on Figure 2e above.

- "The ungrounding of a pinning point..." Are you talking about grounded ice here ? Otherwise I don't understand how the unpinning of a pinning point will provide new warm water pathways, since this is mainly controlled by the bathymetry itself. The unpinning can however decrease the buttressing but has no effect on the warm water pathways. If authors have other thoughts in minds, please provide more details.

Response:

The ungrounding of the pinning point causes a speed-up of the ice, changing the base slope and enhancing ocean circulation, this feedback is described in the cited reference i.e. Holland et al., 2023. We modified the text as follows: "We note that changes in geometry led to enhanced basal melt under Thwaites via "geometrical feedback"³³, particularly due to the onset of melting under the areas newly exposed to the ocean and due to changes in ice basal slope"

- First, note that it is possible to have grounding line retreat on prograde slopes too in case of instability (Hill et al., 2023). Furthermore, recent work from Rignot et al., 2024 have pointed toward significant grounding line modulation specifically in the Milillo cavity (up to 5 km inland). In this region, the authors have shown that irregular warm water intrusions are penetrating up to 6 km inland beneath grounded ice. Such phenomena could challenge your proposed explanation, given the uncertainty on the timing of lake drainage. Please discuss this point.

Response:

Agreed, grounding line retreat can occur on prograde slopes under certain conditions, that of instability being one. And indeed we explicitly discuss the fact that Thwaites grounded ice is showing the highest degree of sensitivity to changes in grounding line and basal melt rate at the very location where the various events described in our study are taking place. We have further developed the concept of instability in the revised manuscript as follows: "We examine in Figure 1c a proxy for spatially resolved ice-shelf buttressing: the sensitivity of grounded ice volume change to the pattern of melt^{63,65,66} (see Methods). The result shows that the largest impacts of melt under floating ice is in the TWITgl region. The result agrees qualitatively with a different buttressing proxy based on the ice-shelf stress state⁶⁴, and together with the even stronger response under the 2013 ungrounding region, suggests that any increases to melt rates in this region will lead to a strong response in ice-stream thinning and acceleration (Figure 5.)^{63,64}."

Indeed the mechanism of seawater intrusion is a potentially key process at the grounding line, and similarly to the observations discussed in our manuscript, highlight the fact that while grounding lines are of particular importance for ice sheet stability, they are still poorly observed, understood, and modeled. As such, the study by Rignot et al., 2024, and our results, are very complementary, shedding light on grounding processes and the role of the ocean and of the subglacial system in modulating ocean melting of the ice sheet. We added a dedicated paragraph to the discussion section on impact as follows:

"These new observations highlight how little we still know about processes taking place at the grounding zone, despite its key importance for ice sheet stability. Amongst these processes, there is a growing body of evidence suggesting that seawater intrusions are taking place along Antarctic grounding lines, which may enhance the ability of the ocean to melt the ice sheet⁷¹. It is still unclear how these intrusions can interact with the subglacial system and the process described in our study. These two processes operate on different time scales, with seawater intrusions taking place via tidal modulations while subglacial lake discharge causing pulses of freshwater release at much lower temporal frequency. We can hypothesise that both processes interact, for example through cavities formed by seawater intrusions creating weakly grounded regions, enhancing the ability of subglacial discharge to cause sudden melt-driven ice retreat. Conversely, exceptional subglacial discharge could enhance seawater intrusion by flushing the grounding zone and melting out a wide sub-glacial connection that is then flooded by seawater. The process described here and the observation of seawater intrusions both challenge our current representation of grounding lines in the models used to forecast future ice loss."

- "Implication for stability of Thwaites". It is not clear whether you are able or not to reproduce changes in melting rates below the ice shelves with the addition of freshwater.

AND

- Can you more clearly express if you are able to reproduce the changes observed in runoff and basal melt rates using the model? I think you have all the tools to do that. Using a simplified approach, as in Rignot et al., (2016), which model submarine melting as a function of increased ocean forcing and runoff, could also be a good and more direct point of comparison. Please provide a scatterplot or comparison of modelled vs observed melting rates using the measured freshwater discharge.

Response:

We show below in a variation of figure 2e the comparison between the basal melt timeseries obtained from observations and that of the ocean model using the observed subglacial flux as input. Any quantitative agreement or disagreement is to be taken with extreme caution here as i) the grounding line region geometry is poorly resolved considering the complex processes acting there (Rignot et al, 2024), ii) the location of the subglacial outflow is somewhat uncertain, iii) model parameterisations for melting and mixing are highly uncertain (see Holland et al., 2023), iv) the ocean temperature in this region is poorly known. For this reason we prefer not to provide a quantitative comparison in the manuscript. The main point of the model in this study is to propose the qualitative power-law behaviour which we are much more confident makes sense. This power-law relationship is also seen, we believe for the first time under ice shelves, and once previously at tidewater glaciers, in observations (see new figure 3b). We now expand on this in the manuscript in the section on the links between basal melt pulse, lake discharge, and grounding line thinning and retreat.

This power law was also used in parameterisations by Xu et al., 2013 and Rignot et al., 2016. Our initial manuscript made reference to the study by Xu et al., 2013, we also added the reference to Rignot et al., (2016), both as context for the impact of subglacial runoff on submarine melting, and in the discussion of the results.

Methods section

- How much swath vs POCA was used? What is the impact of the large imprint of POCA on the resolution of basal melting rates? How much is the CryoSat-2 penetration ? Does it evolve with elevation/snow conditions?

Response:

The ratio of POCA versus SWATH measurements is 1 to 220, we added this information in the methods section. POCA is used to minimize the impact of climate on radar penetration, hence having impact on kilometer scale change, as such the relative lower resolution of the corrective POCA has little impact on the basal melt resolution, this approach was applied and described in Smith et al., 2017. We added further details to the methods section as follows:

“The ratio of the number of swath measurements versus POCA measurements is 220 to 1, differences between swath and POCA elevation are low-pass filtered (100 km) to extract the large scale volume scattering signal, the difference is then subtracted from the full resolution swath dataset.”

Smith, B. E., Gourmelen, N., Huth, A., and Joughin, I.: Connected subglacial lake drainage beneath Thwaites Glacier, West Antarctica, *The Cryosphere*, 11, 451–467, <https://doi.org/10.5194/tc-11-451-2017>, 2017.

- Please provide a code repository for CryoSat-2 processing as it is central to the paper findings, and to comply with open science/FAIR principles

Response:

We now provide a repository for the CryoSat-2 processing, and update the code availability section.

<https://git.ecdf.ed.ac.uk/cryosphere/thw-sgl-mlt>

- Basal melting rates: Provide the data used to derive the results of the study (time series of basal melting rates). It is critical to verify the analysis and to comply with open science/fair principles

Response:

The data repository now contains data necessary for deriving the basal melt timeseries, that is the mean basal melt map, as well as the time-dependent terms described in the methods section.

- Basal melting rates: Provide details on how this was calculated. Which ice velocities product, did changes were monitored in lagrangian framework? If not, the final product of basal melting rates could be much more aliased (see Shean et al., 2019). Also, which firn air content products? Provide details for what you call "standard approaches". You can't expect people to know everything, the methodology should be reproducible, and it is not the case right now. What are the uncertainties on firn air content and SMB?

Response:

We revised the method section to provide the required information:

"We use mass conservation to derive ice shelf mean basal melt rate between 2010 and 2021, and basal melt time-series between 2011 and 2021, by combining measures of elevation and elevation change, Surface Mass Balance, firn air content, and ice velocity^{8,73,74,78}. Melt time-series excludes data from 2010 and early 2011 due to more limited radar altimetry coverage. We update results obtained in ref. ⁷⁴ over Thwaites by using ice velocities for year 2015 from ITS_LIVE⁷⁹ as opposed to the ITS_LIVE composite as previously used. This allows retrieval of melt rate over a larger portion of the TWIT sector, impacted by fracture development in the ITS_LIVE composite. To calculate the melt time-series we consider variation in surface elevation, Surface Mass Balance and firn air content from the RACMO and FDM models^{80,81} (Figure S3)."

The impact of varying velocity is indeed described in our supplementary material, section 13. We note that Shean et al. generated high spatial resolution maps of basal melt rate, but low temporal resolution basal melt change e.g. the effective resolution of the change in velocity was 610 days. Their approach differs considerably from that of e.g. Adusumili et al., 2020, which we apply here, seeking to retrieve high temporal resolution but relatively lower spatial resolution (i.e. sector-wide). We also compare our melt results with a recently published high-resolution melt study of Thwaites based on REMA DEMs (ref. 40 i.e. Chartrand et al., 2024), which follows a similar approach to Shean et al., and found close agreement. We added the following to the main text and Methods section:

Main:

"Our melt rates are in close agreement with a recent high-resolution study⁴⁰ (see Methods)."

Methods:

"Our melt rates agree well with a recent high-resolution melt study⁴⁰, mean basal melt rate of overlapping measurements is 20.4 m yr⁻¹ for this study, and 21.8m yr⁻¹ and 21.6 m yr⁻¹ for ref.⁴⁰'s 2011-2015 and 2016-2019 periods respectively. Ref. ⁴⁰ also found relatively higher rate of mean basal melting under TWITgl during a period including the basal melt pulse discussed in this study i.e. 66.9 m yr⁻¹ (2011-2015) compared with 52.5 m yr⁻¹ (2016-2019) and 55.7 m yr⁻¹ (2020-2023)."

I am reminding below the guidelines for data sharing in nature journals.

A condition of publication in a Nature Portfolio journal is that authors are required to make materials, data, code, and associated protocols promptly available to readers without undue qualifications.

All published manuscripts reporting original research in Nature Portfolio journals must include a data availability statement. The data availability statement must make the conditions of access to the “minimum dataset” that are necessary to interpret, verify and extend the research in the article, transparent to readers

Authors must make available upon request, to editors and reviewers, any previously unreported custom computer code or algorithm used to generate results that are reported in the paper and central to its main claims. Any reason that would preclude the need for code or algorithm sharing will be evaluated by the editors who reserve the right to decline the paper if important code is unavailable.

Availability and peer review of computer code and algorithm

Authors must make available upon request, to editors and reviewers, any previously unreported custom computer code or algorithm used to generate results that are reported in the paper and central to its main claims. Any reason that would preclude the need for code or algorithm sharing will be evaluated by the editors who reserve the right to decline the paper if important code is unavailable.

For all studies using custom code or mathematical algorithm that is deemed central to the conclusions, a statement must be included under the heading "Code availability", indicating whether and how the code or algorithm can be accessed, including any restrictions to access. Code availability statements should be provided as a separate section after the data availability statement but before the references

Response:

Apologies for this oversight in our initial submission, we now provide the relevant data and code and describe them in the data and code availability sections.

REVIEWERS' COMMENTS

Reviewer #2 (Remarks to the Author):

Thanks to the authors for taking the time to respond to all the points. I find the flow of ideas is much improved. And I appreciate the sketch of the system. My only comment at this stage is that the figures could perhaps be improved in places so that the fonts are all readable at sensible figure sizes.

Response:

Thank you. We have increased the font size in the new figures where possible.

Reviewer #3 (Remarks to the Author):

Reviewer #4 (Remarks to the Author):

In general, the comments raised in the previous version of the paper have been well addressed. This has greatly improved the quality and scientific rigor of the paper. From a scientific perspective, I only have a few additional minor comments to add, which can be found below.

Response:

Thank you.

Regarding data availability, I thank the authors for making an effort to share the code. A Google Drive link has also been provided with access to certain types of data. I would like to draw special attention to the high necessity of providing an appropriate and functional DOI before the paper's publication, once again to comply with the policy of Nature journals. Moreover, the repository only provides thickness change data, which represents only a small portion of the dataset used to conduct the scientific analysis of this study. One particularly critical piece of data is missing: basal melting. This is essential and must be included in the dataset. The absence of this specific and distributed cartographic information (in raster or NetCDF format) is critical, as it prevents users from replicating the analysis conducted in this study, hence verifying if the results can be reproduced.

Response:

We have uploaded the dataset to a Zenodo repository and create a DOI that is now provided in the Data Availability statement.

The repository also contains basal melting dataset, in addition to thickness change data, these are the files:

2e_Pine_Island_basal_melt_anomalies.csv

2e_Thwaites_basal_melt_anomalies.csv

SI_melt_components_TWITgl.csv

SI_melt_components_Thwaites.csv

SI_melt_components_TEIS.csv

1b_basal_melt.tif (raster)

The repository contains all the information to replicate the analysis conducted in our study.

If this is addressed (along with minor comments below), the paper will be ready published, and is a great contribution to our understanding of processing affecting ice shelf melting in Antarctica.

Response:

Thank you for your assessment.

Minor comments.

Figure S2. You should still add the areas where basal melting was calculated.

Response:

We modify the caption as follows:

“Basal melt time-series for Thwaites and Pine Island ice shelves (see figure 1 for location and extent of sectors). “Thwaites” corresponds to the sector including both TEIS and TWITgl.”

Suggestion of figure: Providing a movie of changes in basal melting rates spatially through time would be great in illustrating the pulse in basal melting rates.

Response:

We agree that this would be a valuable animation, however we only derive basal melt rate timeseries at the scale of entire ice shelf sectors, spatially detailed basal melt maps are not part of the processing chain and outputs.

Fig 2e. What is the point of showing the evolution of PIG on this graph? The different evolution of the melting could also be related to the different geomorphological setup of the two glaciers (e.g. bathymetry...).

Response:

The reviewer is correct, several factors can affect the way ocean heat content impact melting under ice shelves. We find however that basal melting under PIG is a useful information to provide as these two sectors are in principle impacted by the same ocean variability, and also because the moorings in support of the discussion on ocean conditions are located at the front of PIG. Note that the discussion makes mention of the complexity of the relationship:

L126: “This lack of correlation suggests that the relationship between melting and OHC is not straightforward, or is not well represented between our moorings and satellite melt derivation (Methods). Geometric feedback and ocean circulation, on the continental shelf but also at the ice front and inside the cavity, or strong ocean stratification, can modify this relationship^{31,33,47}”

Hence we think it is important to show the evolution of PIG also in this figure.

Fig S6. Could the lower melting observed at TEIS be explained by the separation of the part of the ice shelves by a ridge in the bathymetry, which could have prevented enhanced basal melting rates?

Response:

The ridge certainly plays a part in shaping the ocean circulation and ocean melting under the Thwaites cavity. It is likely that the ridge, the general westerly ocean circulation in front of the PIG-Thwaites system, the distance from the subglacial outflow, and the depth of the ice draft all play a part in the pattern of basal melting observed under TEIS and TWIT.

L339-340. Concerning the swath measurements, how much swath is used on the ice shelf? Don't you have phase ambiguity issues due to low surface slopes?

Response:

The reviewer is right that swath processing relies on the presence of a surface slope to be viable, for very low slopes a left-right ambiguity can occur. This usually results in degraded signal quality e.g. low coherence, which can be identified and removed from the dataset, and incoherent phase values for a given waveform. Over ice shelves we found through various previous studies (Gourmelen et al., 2017; Wuite et al., 2019; Lilien et al., 2019; Goldberg et al., 2019; Wei et al., 2020; Davison et al., 2023; Gwyther et al., 2023; Surawy-Stepney et al., 2023) that we can retrieve swath elevation for a significant proportion of CryoSat waveforms over ice shelves, we hypothesize that it has to do with ice shelf topography, heterogeneity in backscattering properties across ice shelves, and to the fact that CryoSat

is not pointing perfectly at nadir but is mispointing slightly (Recchia et al., 2017). As a result, we have ~220 times more usable measurements from swath processing than from conventional POCA processing.